# AFlow:
# Automating Agentic Workflow Generation

**Jiayi Zhang**[1,2]*, **Jinyu Xiang**[1]*, **Zhaoyang Yu**[3], **Fengwei Teng**[3], **Xiong-Hui Chen**[4],
**Jiaqi Chen**[5], **Mingchen Zhuge**[6], **Xin Cheng**[3], **Sirui Hong**[1], **Jinlin Wang**[1], **Bingnan Zheng**[5],
**Bang Liu**[7], **Yuyu Luo**[2,8]†, **Chenglin Wu**[1]†

[1]DeepWisdom, [2]The Hong Kong University of Science and Technology (Guangzhou),
[3]Renmin University of China, [4]Nanjing University, [5]Fudan University,
[6]King Abdullah University of Science and Technology, [7]Université de Montréal & Mila,
[8]The Hong Kong University of Science and Technology

## Abstract

Large language models (LLMs) have demonstrated remarkable potential in solving complex tasks across diverse domains, typically by employing agentic workflows that follow detailed instructions and operational sequences. However, constructing these workflows requires significant human effort, limiting scalability and generalizability. Recent research has sought to automate the generation and optimization of these workflows, but existing methods still rely on initial manual setup and fall short of achieving fully automated and effective workflow generation. To address this challenge, we reformulate workflow optimization as a search problem over code-represented workflows, where LLM-invoking nodes are connected by edges. We introduce **AFlow**, an automated framework that efficiently explores this space using Monte Carlo Tree Search, iteratively refining workflows through code modification, tree-structured experience, and execution feedback. Empirical evaluations across six benchmark datasets demonstrate AFlow's efficacy, yielding a 5.7% average improvement over state-of-the-art baselines. Furthermore, AFlow enables smaller models to outperform GPT-4o on specific tasks at 4.55% of its inference cost in dollars. The code is available at https://github.com/FoundationAgents/AFlow.

## 1 Introduction

Large Language Models (LLMs) have emerged as powerful tools for solving complex tasks across various domains, including code generation, data analysis, decision-making, and question answering (Liu et al., 2024; Li et al., 2024a; Zhu et al., 2024; Xie et al., 2024b; Sun et al., 2024; Wang et al., 2024b; Song et al., 2023; Xie et al., 2024a; Zhong et al., 2024a). However, the rapid advancement of LLMs heavily relies on manually designed agentic workflows – structured sequences of LLM invocations accompanied by detailed instructions. Designing and refining these workflows requires significant human effort, which limits the scalability and adaptability of LLMs to new, complex domains and hinders their ability to transfer skills across diverse tasks (Tang et al., 2024).

Recent efforts have focused on automating the discovery of effective agentic workflows to reduce the reliance on human intervention (Khattab et al., 2024; Yüksekgönül et al., 2024; Liu et al., 2023; Hu et al., 2024). Despite these advancements, full automation has not been achieved. For instance, Khattab et al. (2024) requires manual workflow setup before automated prompt optimization. Similarly, methods proposed by Yüksekgönül et al. (2024) and Zhuge et al. (2024) fail to capture the full diversity of workflows necessary for a wide range of tasks (Yu et al., 2023; Yang et al., 2024b; Sun et al., 2023), as their optimization objectives struggle to represent the breadth of possible workflows. The inability to effectively model diverse workflow structures within these automated systems limits their utility and impact. ADAS (Hu et al., 2024) represents workflows using code, achieving a

---

*These authors contributed equally to this work.

†Corresponding authors: Yuyu Luo (E-mail:yuyuluo@hkust-gz.edu.cn), Chenglin Wu (E-mail: alexanderwu@deepwisdom.ai)

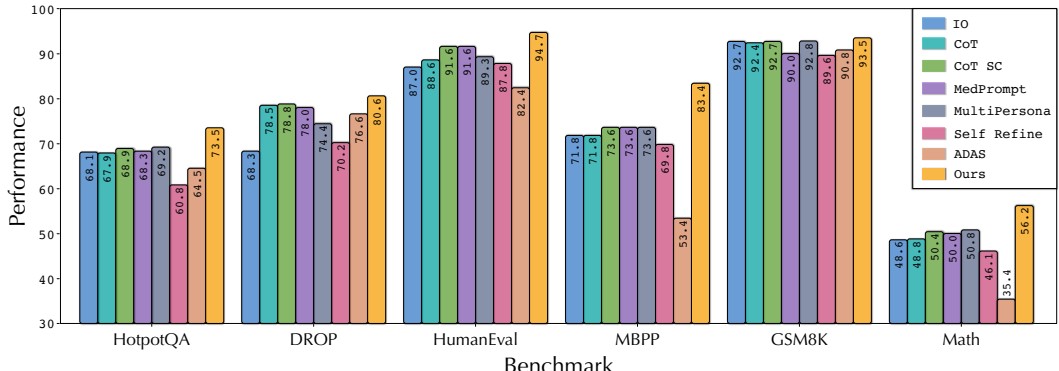

Figure 1: **Performance comparison with other methods.** To assess the method's performance, we employ various metrics across different datasets: solve rate for Math and GSM8K, F1 score for HotpotQA and DROP, and pass@1 for HumanEval and MBPP. Our AFLOW (highlighted in yellow) consistently outperforms all automated workflow optimization and manually designed methods across all six benchmarks.

relatively complete representation. However, due to the efficiency limitations of its linear heuristic search algorithm, ADAS struggles to generate effective workflows within a limited number of iterations. This highlights the need for more effective techniques to represent and automate the generation of agentic workflows, which would accelerate the application of LLMs across domains.

In response to these challenges, we introduce an innovative framework for automatically generating agentic workflows. Our **key idea** is to model the workflow as a series of interconnected LLM-invoking nodes, where each node represents an LLM action and the edges define the logic, dependencies, and flow between these actions. This structure transforms the workflow into a vast search space, encompassing a wide variety of potential configurations. Our goal is to efficiently navigate this space, automatically generating optimized workflows that maximize task performance while minimizing human intervention.

However, the diversity and complexity of tasks present significant challenges. Specifically, each task can have different requirements, operations, and dependencies, which makes it difficult to represent them in a unified yet flexible manner (Chen et al., 2021; Cobbe et al., 2021; Yang et al., 2018; Luo et al., 2018). Furthermore, the search space for possible workflows, comprising an immense number of code structures and node configurations, is virtually boundless, creating an additional challenge for efficient exploration and optimization.

To address these challenges, we propose AFLOW, a Monte Carlo Tree Search (MCTS)-based framework designed to systematically explore and discover optimal agentic workflows. AFLOW represents workflows as flexible nodes connected by code-based edges, which encapsulate possible relationships such as logical flows, conditions, and dependencies. These edges allow the workflow to be modeled as a graph (Zhuge et al., 2024) or network (Liu et al., 2023), offering a powerful structure for capturing complex interactions between LLM invocations.

To enhance the search process and improve efficiency, AFLOW introduces a novel concept of operators – predefined, reusable combinations of nodes representing common agentic operations (e.g., Ensemble, Review & Revise). These operators serve as foundational building blocks for constructing workflows and are integrated into the search space, ensuring that the exploration process leverages known patterns of effective agentic operations.

AFLOW employs the MCTS algorithm to navigate this infinite search space. The framework's workflow optimization process incorporates several key innovations: a soft mixed-probability selection mechanism for node exploration, LLM-driven node expansion to introduce new possibilities, execution evaluation to assess workflow performance, and backpropagation of experience to refine future search iterations. This combination of techniques ensures that AFLOW efficiently discovers workflows that adapt to the complexity of diverse tasks while reducing reliance on manual intervention.

We make the following key contributions: (1) **Problem Formulation**: We formalize the workflow optimization problem, generalizing prior approaches as specific cases. This provides a unified framework for future research at both the node and workflow optimization levels. (2) **AFLOW**:

We introduce AFLOW, an MCTS-based method that automatically discovers effective workflows across multiple domains with minimal human intervention. (3) **Extensive Evaluation**: We evaluate AFLOW on six benchmark datasets: HumanEval, MBPP, MATH, GSM8K, HotPotQA, and DROP. AFLOW outperforms manually designed methods by 5.7% and surpasses existing automated approaches by 19.5%. Notably, workflows generated by AFLOW enable smaller LLMs to outperform larger models, offering better cost-performance efficiency, with significant implications for real-world applications.

## 2 RELATED WORK

**Agentic Workflow**  Agentic workflow and autonomous agents  (Zhuge et al., 2023; Hong et al., 2024a; Zhang et al., 2024c; Wang et al., 2023; Liu et al., 2025) represent two distinct paradigms of LLM application. The former completes tasks statically through predefined processes with multiple LLM invocations, while the latter solves problems dynamically through flexible autonomous decision-making. Compared to autonomous agents that require specific actions and decision patterns designed for the environment, agentic workflows can be constructed based on existing human domain experience and iterative refinement, offering higher potential for automated construction.

Agentic workflows can be broadly categorized into general and domain-specific types. General workflows emphasize universal problem-solving approaches, such as  (Wei et al., 2022; Wang et al., 2022; Madaan et al., 2023; Wang et al., 2024a; Teng et al., 2025). Domain-specific workflows focus on building effective processes to solve domain-specific problems, such as code generation  (Hong et al., 2024b; Ridnik et al., 2024; Zhong et al., 2024a), data analysis  (Xie et al., 2024b; Ye et al., 2024; Li et al., 2024a; Zhou et al., 2023), mathematics  (Zhong et al., 2024b; Xu et al., 2024), question answering  (Nori et al., 2023; Zhou et al., 2024a). Existing work has manually discovered numerous effective agentic workflows, but it's challenging to exhaust various tasks across different domains, further highlighting the importance of automated workflow generation and optimization.

**Automated Agentic Optimization**  Recent work aims to automate the design of agentic workflows, categorized into three types: automated prompt optimization, hyperparameter optimization, and automated workflow optimization. Prompt optimization  (Fernando et al., 2024; Yüksekgönül et al., 2024; Yang et al., 2024a; Khattab et al., 2024; Xiang et al., 2025) uses LLMs to optimize prompts within fixed workflows. Hyperparameter optimization (Saad-Falcon et al., 2024) focuses on optimizing predefined parameters. While these approaches improve performance, they are limited in generalization to new tasks and often require moderate human effort for task-specific designs.

Automated workflow optimization (Li et al., 2024b; Zhou et al., 2024b; Zhuge et al., 2024; Hu et al., 2024) aims to optimize entire workflow structures, offering more potential for fully automated generation. Recent works explore diverse representations and methods. GPTSwarm (Zhuge et al., 2024) uses graph structures with reinforcement learning, but struggles to represent workflows with conditional states due to graph structure limitations. ADAS (Hu et al., 2024) utilizes code structures to represent workflows and stores historical workflows in a linear list structure, aligning closely with our goals. However, it is constrained by the efficiency of its search algorithm as it relies on overly simplistic representations of experiences in the searching process, making it challenging to discover effective workflows.

AFLOW also uses code to represent workflows, but goes further by providing a more fundamental structure called named node. This structure encompasses various LLM invocation parameters, allowing for more detailed workflow representation. We also introduce operators that implement predefined node combination functions. Simultaneously, AFLOW employs a specially designed MCTS algorithm for automated workflow optimization, leveraging the tree-structured experience and execution feedback to efficiently discover effective workflows.

## 3 PRELIMINARY

In this section, we will first formulate the automated agentic workflows generation problem in Section 3.1 and then discuss design considerations of our AFLOW in Section 3.2. For the core concept of this section, we provide an example explanation in Figure 2.

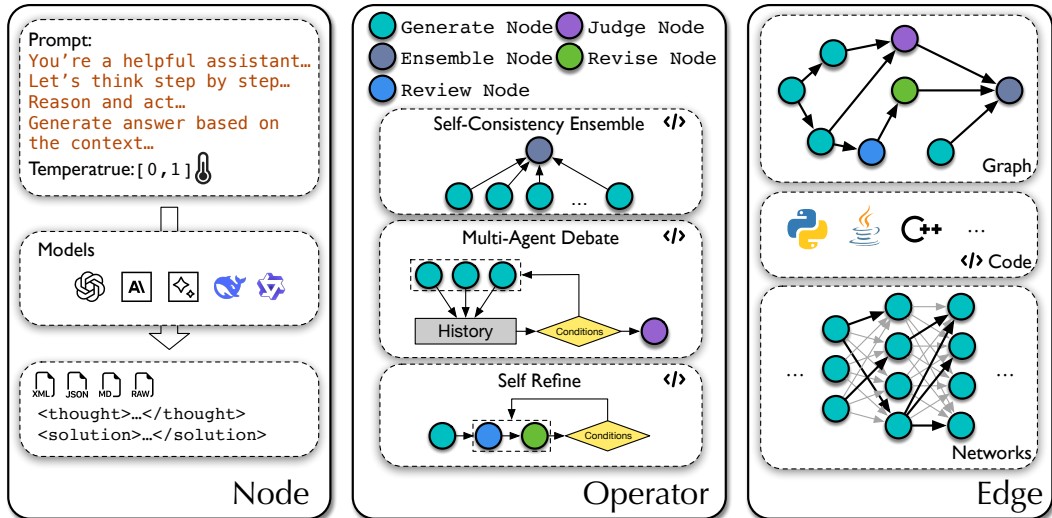

Figure 2: **The example of node, operator, and edge.** We demonstrate the optional parameters for Nodes, the structure of some Operators, and common representations of Edges.

## 3.1 PROBLEM FORMULATION

**Agentic Workflow**  We define an agentic workflow W as a series of LLM-invoking nodes connected by edges to define the exection orders, denoted as $\mathcal{N} = \{N_1, N_2, \ldots, N_i \ldots\}$. Each node $N_i$ represents a specific operation performed by an LLM and is characterized by the following parameters. The code abstraction of the node is shown in Appendix A.2.

- **Model** $M$: The specific language model invoked at node $N_i$.
- **Prompt** $P$: The input or task description provided to the model at each node.
- **Temperature** $\tau$: A parameter controlling the randomness of the LLM's output at node $N_i$.
- **Output format** $F$: The format in which the model's output is structured (*e.g.*, xml, json, markdown, raw). The node in workflow should provide different output formats, inspired by the  Tam et al. (2024).

Edge $E$ represent abstract structures defining node relationships, governing the sequence of execution. The edge $E$ can be represented via various structures, such as:

- **Graph  Zhuge et al. (2024)**: A flexible structure representing hierarchical, sequential, or parallel relationships between nodes, allowing for complex branching workflows.
- **Neural Network  (Liu et al., 2023)**: A structure that can represent complex, non-linear relationships between nodes, allowing for adaptive and learnable workflows based on input and feedback.
- **Code  (Hu et al., 2024)**: A comprehensive representation that can express linear sequences, conditional logic, loops, and incorporate graph or network structures, offering the most precise control over workflow execution for LLMs.

While graph structures can represent workflow relationships, they require complex extensions (e.g., Petri nets, BPMN) beyond basic DAGs to naturally express parallel execution and conditional logic. Neural networks enable adaptive transitions but lack precise control over workflow execution. In contrast, code representation inherently supports all these relationships through standard programming constructs. Therefore, we adopt code as our primary edge structure to maximize expressivity.

**Automated Workflow Optimization**  Given a task $T$ and an evaluation function $G$, the goal of workflow optimization is to discover a workflow $W$ that maximizes $G(W, T)$. This can be formulated as a search process where an algorithm $A$ explores the search space $\mathcal{S}$ to determine the optimal workflow configuration. The search space $\mathcal{S}$ for a workflow optimization problem encompasses all possible configurations of node parameters and edge structures:

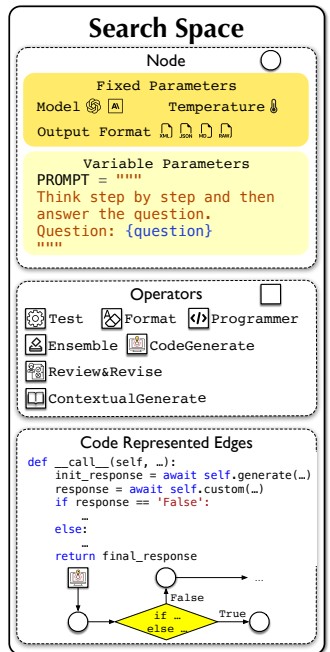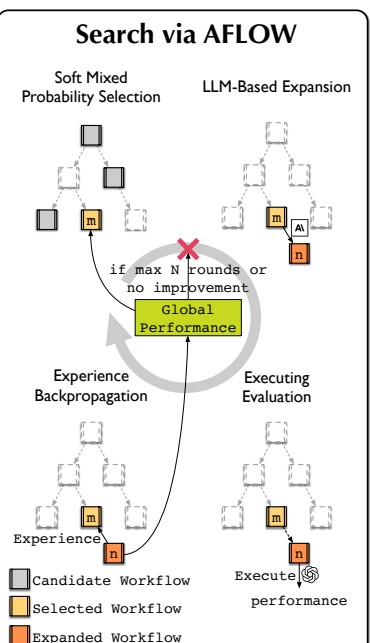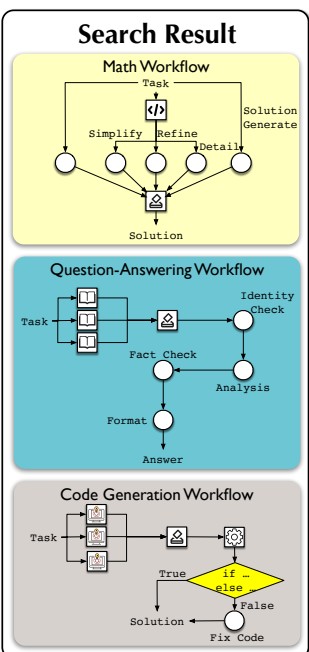

Figure 3: **Overall AFLOW framework**: By setting a search space composed of nodes with only prompt parameters flexible, a given operator set, and a code representing edge, AFLOW performs an MCTS-based search within this space. Through a variant of MCTS designed for workflow optimization, AFLOW iteratively executes a cycle of Soft Mixed Probability Selection, LLM-Based Expansion, Execution Evaluation, and Experience Backpropagation until reaching the maximum number of iterations or meeting convergence criteria.

$$\mathcal{S} = \{(\mathcal{N}, E) \mid E \in \mathcal{E}\},$$

where $\mathcal{N} = \{N(M, \tau, P, F) \mid M \in \mathcal{M}, \tau \in [0, 1], P \in \mathcal{P}, F \in \mathcal{F}\}$, with $\mathcal{M}, \mathcal{P}, \mathcal{F}, \mathcal{E}$ representing the sets of possible language models, prompts, output formats, and edge configurations, respectively.

With this formulation, the workflow optimization problem can be expressed as:

$$W = A(\mathcal{S}, G, T),$$
$$W^* = \arg\max_{W \in \mathcal{S}} G(W, T),$$

where $A$ is the search algorithm that explores the search space $\mathcal{S}$, and $W^*$ is the optimal workflow configuration that maximizes the evaluation function $G$ for the given task $T$.

### 3.2 AFLOW OVERVIEW

**Limitations of Previous Methods** Previous approaches Yüksekgönül et al. (2024); Khattab et al. (2024); Zhuge et al. (2024) to workflow optimization have primarily been constrained by the limited scope of their search spaces, based on problem definition in Section 3.1. Another related work, ADAS (Hu et al., 2024), searches in a larger space comprising a combination of prompts $N(P, T)$ and edges $E$, but fails to discover effective workflows due to the efficiency limitations of its linear heuristic search algorithm.

**Formulation** To address the limitations of previous methods, we propose AFLOW, a novel framework that leverages Large Language Models (LLMs) as optimizers within a variant of Monte Carlo Tree Search (MCTS) to search for optimal workflows. As discussed in Section 3.1, edges can be represented in both graphs and code. To ensure AFLOW can explore the full range of possible agentic workflows, we represent nodes N and edges E through code. Specifically, as shown in Figure 3,

AFLOW uses a variant of MCTS to iteratively explore the workflow search space, evaluate different configurations, and backpropagate experiences to refine the workflow optimization process.

To enhance search efficiency in practice, we simplify the search space by fixing key parameters such as the model $M$, temperature $\tau$, and format $F$. This simplification allows AFLOW to focus its search primarily on the code-represented edges $E$ and prompts. To navigate this still vast search space effectively, we introduce the concept of **Operators**. These Operators encapsulate common agentic operations (e.g., Ensemble, Review, Revise) by combining $N$ and $E$ into unified interfaces, thereby enabling more efficient utilization by AFLOW. By employing these Operators, we achieve more efficient search and streamlined workflow generation.

Formally, given a set of Operators $\mathcal{O}$ that represents predefined node combinations, and an edge space $\mathcal{E}$ represented through code, the optimization problem can be formalized as:

$$\mathcal{S}_{\text{AFlow}} = \{(P_1, \ldots, P_n, E, O_1, \ldots, O_n) \mid P_i \in \mathcal{P}, E \in \mathcal{E}, O_i \in \mathcal{O}\} \tag{1}$$

$$W^* = \text{AFLOW}(\mathcal{S}_{\text{AFlow}}, G, T) \tag{2}$$

**Tasks Scope and Operations** In this paper, we focus on applying AFLOW to reasoning tasks with numerical evaluation functions. We extract common operations from existing literature and define them as part of the operator set $\mathcal{O}$. These operations include: (1) Generate, (2) Format, (3) Review and Revise Madaan et al. (2023), (4) Ensemble Wang et al. (2022), (5) Test Zhong et al. (2024a), (6) Programmer, and (7) Custom as the default operator for basic node construction. The operator set $\mathcal{O}$ can be easily expanded to enhance search efficiency for various tasks. Even without any predefined operators, AFLOW can construct different workflow nodes using the basic Custom operator. The efficiency comparison between these approaches is detailed in Section 5.2. For a comprehensive understanding of the operators, we provide their detailed structures in Appendix A.4.

## 4 The Design Details of AFLOW

The core concept of AFLOW is to employ Large Language Models (LLMs) as optimizers within a Monte Carlo Tree Search (MCTS) variant to discover effective workflows. In our MCTS structure, **each tree node represents a complete workflow rather than individual LLM-invoking node**, enabling the discovery of universal solutions for classes of problems. The search process operates through an iterative cycle of soft mixed probability selection, LLM-based optimization expansion, execution evaluation, and experience backpropagation until reaching maximum iterations or convergence criteria. A simplified illustration is shown in Figure 3, with detailed algorithm process and theoretical analysis presented in Appendix A.6 and Appendix G, respectively.

Existing workflow optimization methods iteratively use past workflow structures to prompt LLMs to discover new structures. However, due to information loss during accumulation (as input tokens increase), this approach struggles to guide LLMs towards specific performance metrics. Combined with the vast search space of code, this reduces search efficiency. Our **key idea** is to leverage the tree structure of MCTS to preserve workflow-based exploration experiences in $N_{max}$ rounds workflow optimization. When a workflow is revisited, we accurately reuse past successful experiences and avoid failures, enabling effective workflow generation and improving search efficiency. To prevent local optima, we introduce a special selection mechanism allowing generation from a blank template at any round. Next, we will introduce the complete process of AFLOW, as shown in Algorithm 1.

**Initialization** AFLOW begins with a template workflow $W_0$, which provides a framework for invoking nodes and operators. The code template, detailed in Appendix A.3, allows the LLM optimizer to complete workflow simply by completing call functions. Prior to initiating the search process, we randomly partition the dataset into a validation set (20%) and a test set (80%), with the random seed fixed at 42. To optimize computational efficiency, AFLOW then executes the blank template five times on the validation dataset. From these executions, we select a subset of problems that exhibit high variance in scores, which becomes the final validation set.

**Selection** Our algorithm forms the initial workflow by evaluating an empty workflow on the validation set. And then continuously select workflows based on a soft mixed probability selection strategy. cWe propose this strategy for workflow optimization: combining uniform and score-based

---

**Algorithm 1** Algorithm of AFLOW: Detailed implementation

---

**Require:** Evaluator $G$, Dataset $D$, Operators $\mathcal{O}$
**Ensure:** Optimized Workflow $W^*$
1: Initialize $W_0$, split $D$ into $D_V$ and $D_T$
2: $W^* \leftarrow W_0$
3: **for** $iteration \leftarrow 1$ to $N_{max}$ **do**
4:     $workflow \leftarrow$ Select(tree)             ▷ Using soft mixed probability strategy
5:     $child.workflow \leftarrow$ Expand($workflow, \mathcal{O}$)            ▷ LLM-based expansion
6:     $score \leftarrow$ Evaluate($child.workflow, G, D_V$)         ▷ Multiple runs for robustness
7:     Backpropagate($child.workflow, score$)         ▷ Update experience and scores
8:     Update $W^*$ if improved
9:     **if** ConvergenceCriteriaMet() **then break**
10:     **end if**
11: **end for**
12: **return** $W^*$

---

weighted probability distributions to select from top-k workflows and the initial workflow, where including the initial workflow ensures persistent exploration capability while avoiding local optima. The formula for this selection strategy is as follows:

$$P_{\text{mixed}}(i) = \lambda \cdot \frac{1}{n} + (1-\lambda) \cdot \frac{\exp(\alpha \cdot (s_i - s_{\max}))}{\sum_{j=1}^{n} \exp(\alpha \cdot (s_j - s_{\max}))}, \tag{3}$$

where $n$ is the number of workflows, $s_i$ is workflow $i$'s score, $s_{\max}$ is the maximum score, $\alpha$ (0.4) controls score influence, and $\lambda$ (0.2) balances exploration and exploitation.

**Expansion** In the expansion phase, we employ an LLM as an optimizer to create new workflows and the optimize prompt is illustrated in Appendix A.1. The optimizer leverages the selected workflow's experience to generate new prompts or modify node connections by altering code, resulting in new workflows. Specifically, to maximally uncover insights from past iterations, the experience includes all modifications and their corresponding improvements or failures on the selected workflow, along with precise logs of predictions and expected output.

**Evaluation** AFLOW directly executes workflows to get feedback due to explicit evaluation functions in reasoning tasks. We test each generated workflow 5 times on the validation set, computing mean and standard deviation. While this increases per-iteration cost, it provides more accurate feedback for the optimizer. This precision enhances search efficiency, ultimately reducing the number of iterations required to reach an effective solution.

**Backpropagation** After execution, we record: (1) the workflow's performance, (2) the optimizer's modification of its parent workflow, and (3) optimization success relative to its parent. This information is stored in experience and propagated back to the parent workflow, while the performance score is added to the global record for selection.

**Terminal Condition** We implement early stopping to reduce unnecessary execution costs: the process terminates if the top-k average score shows no improvement for $n$ consecutive rounds, or after $N$ total rounds otherwise. See Appendix A.6 for algorithmic details.

## 5 EXPERIMENTS

### 5.1 EXPERIMENTAL SETUP

**Datasets** We utilized six public benchmarks for our experiments. Following established practices (Saad-Falcon et al., 2024; Hu et al., 2024) in workflow optimization, we divide the data into validation and test sets using a 1:4 ratio. Specifically, we use the full datasets for GSM8K (Cobbe et al., 2021), HumanEval (Chen et al., 2021), and MBPP (Austin et al., 2021). For HotpotQA (Yang et al., 2018) and DROP (Dua et al., 2019), we randomly select 1,000 samples each, in line with (Hu et al., 2024; Shinn et al., 2023). For the MATH (Hendrycks et al., 2021) dataset, we follow (Hong

et al., 2024a) in selecting 617 problems from four typical problem types (Combinatorics & Probability, Number Theory, Pre-algebra, Pre-calculus) at difficulty level 5.

**Baselines** We compare workflow discovered by AFLOW against manually designed methods for LLMs, including IO (direct LLM invocation), Chain-of-Thought (Wei et al., 2022), Self Consistency CoT (5 answers) (Wang et al., 2022), MultiPersona Debate (Wang et al., 2024a), Self-Refine (max 3 iteration rounds) (Madaan et al., 2023), and MedPrompt (3 answers and 5 votes) (Nori et al., 2023). We also compared against workflow designed by automated workflow optimization method ADAS (Hu et al., 2024).

**Implementation Details** AFLOW utilizes different models for optimization and execution. We employ Claude-3.5-sonnet (Anthropic, 2024) as the optimizer and use models: DeepSeek-V2.5 (Deepseek, 2024), GPT-4o-mini-0718 (OpenAI, 2024b), Claude-3.5-sonnet-0620 (Anthropic, 2024), GPT-4o-0513 (OpenAI, 2024a)) as executors. All models are accessed via APIs. We set the temperature to 1 for DeepSeek-V2.5 and to 0 for the other models. We set iteration rounds to 20 for AFLOW. For ADAS, we use Claude-3.5-sonnet as the optimizer and GPT-4o-mini as the executor, with the iteration rounds set to 30.

**Metrics.** For GSM8K and MATH$_{lv5*}$, we report the Solve Rate (%) as the primary metric. For HumanEval and MBPP, we report the pass@1 metric as presented in (Chen et al., 2021) to assess code accuracy. For HotpotQA and DROP, we report the F1 Score. Additionally, for all datasets, we calculate the cost by tracking token usage to construct a pareto front, visually demonstrating the performance-cost trade-offs between different methods.

## 5.2 EXPERIMENTAL RESULTS AND ANALYSIS

**Main Results** The main experimental results, as shown in Table 1, demonstrate the effectiveness of AFLOW. Workflows optimized by AFLOW outperform all manually designed methods by an average of **5.7%** and surpass contemporary automatic workflow optimization work by 19.5%. Across six datasets in QA, Code, and Math domains, AFLOW achieves an average performance of 80.3%, marking the capability and usability of this method. Notably, compared to similar works, AFLOW performed better on more challenging tasks, improving over ADAS on MATH$_{lv5*}$ and MBPP tasks by 57%, showcasing the robustness of the model on complex datasets.

Table 1: Comparison of performance between manually designed methods and workflow generated by automated workflow optimization methods in QA, code, and Math scenarios. All methods are executed with GPT-4o-mini on divided test set, and we tested it three times and reported it on the average.

| Method | Benchmarks | | | | | | Avg. |
|---|---|---|---|---|---|---|---|
| | HotpotQA | DROP | HumanEval | MBPP | GSM8K | MATH | |
| IO (GPT-4o-mini) | 68.1 | 68.3 | 87.0 | 71.8 | 92.7 | 48.6 | 72.8 |
| CoT (Wei et al., 2022) | 67.9 | 78.5 | 88.6 | 71.8 | 92.4 | 48.8 | 74.7 |
| CoT SC (5-shot) (Wang et al., 2022) | 68.9 | 78.8 | 91.6 | 73.6 | 92.7 | 50.4 | 76.0 |
| MedPrompt (Nori et al., 2023) | 68.3 | 78.0 | 91.6 | 73.6 | 90.0 | 50.0 | 75.3 |
| MultiPersona (Wang et al., 2024a) | 69.2 | 74.4 | 89.3 | 73.6 | 92.8 | 50.8 | 75.1 |
| Self Refine (Madaan et al., 2023) | 60.8 | 70.2 | 87.8 | 69.8 | 89.6 | 46.1 | 70.7 |
| ADAS (Hu et al., 2024) | 64.5 | 76.6 | 82.4 | 53.4 | 90.8 | 35.4 | 67.2 |
| Ours | **73.5** | **80.6** | **94.7** | **83.4** | **93.5** | **56.2** | **80.3** |

To explore whether the workflow searched by AFLOW is model-agnostic, we use GPT-4o-mini and DeepSeek-V2.5 as execution LLMs to search effective workflows with different structures, with the results illustrated in Table 2. When applying these workflows to other models, the vast majority demonstrate stronger performance than the baseline, showcasing the generalizability of the workflows discovered by AFLOW. Simultaneously, we observe that the workflow identified using DeepSeek-V2.5 performs notably weaker on GPT-4o-mini compared to the workflow found using GPT-4o-mini itself. This suggests that different language models require different workflows to achieve their optimal performance.

**Cost Analysis** We demonstrate the comparison of performance and cost between the baselines and the top three workflows found by AFLOW using GPT-4o-mini and DeepSeek-V2.5 as execution LLMs. The comparison is made across four models with different capabilities and price points.

Table 2: Comparison of performance between manually designed methods and workflows generated by AFLOW with two executor LLM: GPT-4o-mini ("Ours") and DeepSeek-V2.5 ("Ours*"). All workflows are tested thrice on the humaneval test set, with average results reported. "MP" denotes "MedPrompt" (Nori et al., 2023), and "MPD" denotes "MultiPersona Debate" (Wang et al., 2024a). The results demonstrate that workflows obtained through AFLOW exhibit strong transferability.

| Model | Methods | | | | | | | |
|---|---|---|---|---|---|---|---|---|
| | IO | CoT | CoT SC | MP | MPD | SR | Ours | Ours* |
| GPT-4o-mini | 87.0 | 88.6 | 91.6 | 91.6 | 89.3 | 87.8 | 94.7 | 90.8 |
| DeepSeek-V2.5 | 88.6 | 89.3 | 88.6 | 88.6 | 89.3 | 90.0 | 93.9 | 94.7 |
| GPT-4o | 93.9 | 93.1 | 94.7 | 93.9 | 94.7 | 91.6 | **96.2** | 95.4 |
| Claude-3.5-sonnet | 90.8 | 92.4 | 93.9 | 91.6 | 90.8 | 89.3 | 95.4 | 94.7 |

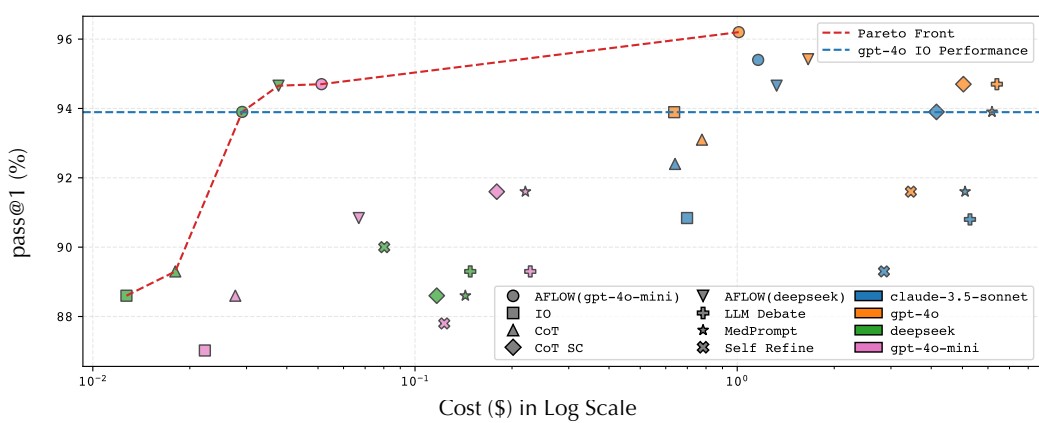

Figure 4: The cost refers to the total expense of executing the divided HumanEval test set. AFLOW (execution model) refers to workflows found by AFLOW using the execution model to obtain feedback. The colors in the legend represent the LLM used to execute each workflow in test dataset. The specific numerical values for this Figure can be found in Appendix D.

Results demonstrate that AFLOW can identify workflows that allow weaker models to outperform stronger models on the pareto front of cost-effectiveness. This breakthrough effectively removes barriers to the widespread application of agentic workflows across various domains. By automating the design of effective agentic workflows, AFLOW eliminates the human labor costs previously required. Moreover, the ability to achieve superior performance at lower costs compared to stronger models opens up further possibilities for widespread adoption.

**Ablation Study** We introduce operators as human-designed effort to enhance search efficiency. An ablation study on GSM8K (Figure 5) shows that operators help AFLOW discover better workflows more efficiently, achieving incremental improvements. Notably, even without operators, AFLOW maintains strong performance (93.1%), surpassing manual designs. Notably, AFLOW autonomously develops ensemble-like structures without operators, demonstrating its capability for independent workflow design and marking a significant step towards full automation. Details is shown in Appendix B.

**Case Study** AFLOW demonstrates a clear iteration process, as shown in Figure 6, illustrating how it evolves from a blank template (containing only a single Node without prompts) to the structure presented in Figure 5(B). In each iteration, AFLOW employs a single-step modification, meaning it either adds one operator (rounds 2, 3) or makes a targeted modification to a prompt (rounds 8, 10). Among the unsuccessful exploration rounds, AFLOW introduced a custom review node that directly modified answers generated through complex processes without additional reasoning (round 5), which decreased accuracy. In round 14, AFLOW attempted to rephrase the problem but overly focused on "discount" information, leading to a decrease in accuracy. This iteration process showcases how tree-based search allows AFLOW to further optimize known paths while retaining

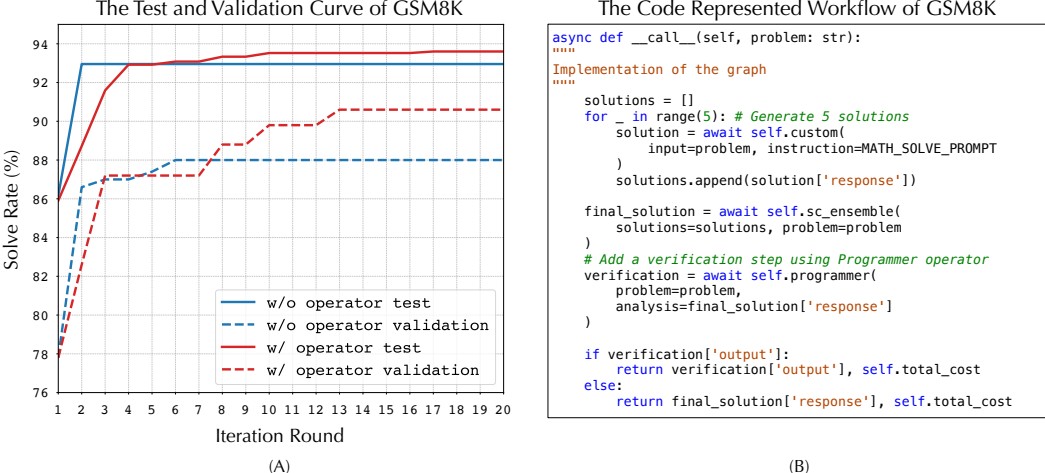

Figure 5: (A) Comparison of highest performance curves on GSM8K for both validation and test sets generated by AFLOW with and without operators. Compared to other datasets, GSM8K has a larger data volume, meaning that the same percentage improvement represents a greater increase in correctly solved samples, avoiding fluctuations in improvement due to small data size that could affect comparisons; (B): The code for the best-performing workflow discovered by AFLOW on the GSM8K dataset.

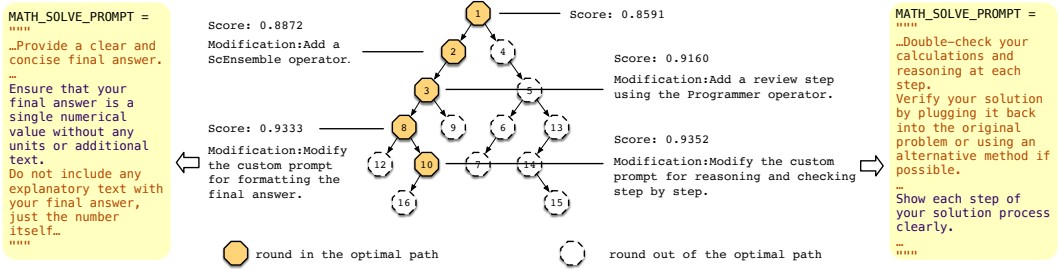

Figure 6: Tree-structured iteration process of AFLOW on GSM8K: We highlight the path from the initial round (round 1) to the best-performing workflow, reporting the score for each node and its modification from the previous node. The purple sections in the prompts on both sides represent the main prompt modifications in this iteration.

the ability to explore new ones. On the MBPP dataset, AFLOW discovered structures similar to current manually designed workflows, such as test generation and execution by LLMs as seen in Ridnik et al. (2024). The workflow and more discovered results are presented in Appendix B and a complete optimization process is presented in Appendix C.

# 6 CONCLUSION

This paper has introduced AFLOW, a novel framework for automated workflow optimization. We have comprehensively formulated the automated workflow optimization problem, establishing a foundational structure for future research. AFLOW has leveraged Monte Carlo Tree Search and code-represented workflows to navigate the vast search space of possible workflows efficiently. Our experiments across six benchmarks demonstrate the effectiveness of AFLOW, which has outperformed manually designed methods and existing automated optimization approaches. Ablation studies have shown that AFLOW can autonomously discover effective structures, even without predefined operators. Importantly, AFLOW has enabled weaker models to outperform stronger ones on the Pareto front of cost-effectiveness. We further discuss the potential applications of AFLOW across diverse domains in Appendix F, potentially revolutionizing the adoption of agentic workflows across various domains. These results have highlighted AFLOW's potential for enhancing LLMs' problem-solving capabilities while optimizing computational costs.

ACKNOWLEDGEMENTS

This paper is supported by NSF of China (62402409), Guangzhou Municipality Big Data Intelligence Key Lab (2023A03J0012), Guangdong Basic and Applied Basic Research Foundation (2023A1515110545), Guangzhou Basic and Applied Basic Research Foundation (2025A04J3935), and Guangzhou-HKUST(GZ) Joint Funding Program (2025A03J3714).

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

# A APPENDIX

## A.1 LLM BASED EXPANSION: PROMPT FOR LLM OPTIMIZER

**Workflow optimize prompt**

```
PROMPT = """You are building a Graph and corresponding Prompt to jointly solve {type}
↪  problems. Referring to the given graph and prompt, which forms a basic example of
↪  a {type} solution approach, please reconstruct and optimize them. You can add,
↪  modify, or delete nodes, parameters, or prompts. Include your single modification
↪  in XML tags in your reply. Ensure they are complete and correct to avoid runtime
↪  failures. When optimizing, you can incorporate critical thinking methods like
↪  review, revise, ensemble (generating multiple answers through different/similar
↪  prompts, then voting/integrating/checking the majority to obtain a final answer),
↪  selfAsk, etc. Consider Python's loops (for, while, list comprehensions),
↪  conditional statements (if-elif-else, ternary operators), or machine learning
↪  techniques (e.g., linear regression, decision trees, neural networks, clustering).
↪  The graph complexity should not exceed 10. Use logical and control flow (IF-ELSE,
↪  loops) for a more enhanced graphical representation.Ensure that all the prompts
↪  required by the current graph from prompt_custom are included.Exclude any other
↪  prompts. Output the modified graph and all the necessary Prompts in prompt_custom
↪  (if needed).The prompt you need to generate is only the one used in
↪  `prompt_custom.XXX` within Custom. Other methods already have built-in prompts and
↪  are prohibited from being generated. Only generate those needed for use in
↪  `prompt_custom`; please remove any unused prompts in prompt_custom. the generated
↪  prompt must not contain any placeholders. Considering information loss, complex
↪  graphs may yield better results, but insufficient information transmission can
↪  omit the solution. It's crucial to include necessary context during the
↪  process."""
```

## A.2 BASIC STRUCTURE OF NODE

**Node structure**

```python
class ActionNode:
    async def fill(self, context, llm, schema...):
        """
        :param context: Everything we should know when filling node.
        :param llm: Large Language Model with pre-defined system message.
        :param schema: json/markdown/xml, determine example and output format.
         - raw: free form text
         - json: it's easy to open source LLM with json format
         - markdown: when generating code, markdown is always better
         - xml: its structured format is advantageous for constraining LLM outputs
        """
        ...
        return self
```

## A.3 BASIC STRUCTURE OF WORKFLOW

**Workflow structure**

```python
DatasetType = Literal["HumanEval", "MBPP", "GSM8K", "MATH", "HotpotQa", "DROP"]

class Workflow:
    def __init__(
        self,
        name: str,
        llm_config,
        dataset: DatasetType,
    ) -> None:
        self.name = name
        self.dataset = dataset
        self.llm = create_llm_instance(llm_config)
        self.llm.cost_manager = CostManager()
```

```python
    async def __call__(self, problem: str):
        """
        Implementation of the workflow
        """
        raise NotImplementedError("This method should be implemented by the subclass")
```

## A.4 OPERATORS

**Operators**

```python
class ContextualGenerate(Operator):
    async def __call__(self, problem, context, mode: str = None):
        prompt = CONTEXTUAL_GENERATE_PROMPT.format(problem_description=problem,
        ↪  thought=context)
        fill_kwargs = {"context": prompt, "llm": self.llm}
        if mode:
            fill_kwargs["mode"] = mode
        node = await ActionNode.from_pydantic(GenerateOp).fill(**fill_kwargs)
        response = node.instruct_content.model_dump()
        return response

class CodeGenerate(Operator):
    async def __call__(self, problem, function_name, mode: str = None):
        prompt = GENERATE_CODEBLOCK_PROMPT.format(problem_description=problem)
        fill_kwargs = {"context": prompt, "llm": self.llm, "function_name":
        ↪  function_name}
        if mode:
            fill_kwargs["mode"] = mode
        node = await ActionNode.from_pydantic(CodeGenerateOp).fill(**fill_kwargs)
        response = node.instruct_content.model_dump()
        return response

class Format(Operator):
    async def __call__(self, problem, solution, mode: str = None):
        prompt = FORMAT_PROMPT.format(problem_description=problem, solution=solution)
        fill_kwargs = {"context": prompt, "llm": self.llm}
        if mode:
            fill_kwargs["mode"] = mode
        node = await ActionNode.from_pydantic(FormatOp).fill(**fill_kwargs)
        response = node.instruct_content.model_dump()
        return response

class Review(Operator):
    async def __call__(self, problem, solution, mode: str = None):
        prompt = REVIEW_PROMPT.format(problem_description=problem, solution=solution,
        ↪  criteria=self.criteria)
        fill_kwargs = {"context": prompt, "llm": self.llm}
        if mode:
            fill_kwargs["mode"] = mode
        node = await ActionNode.from_pydantic(ReviewOp).fill(**fill_kwargs)
        response = node.instruct_content.model_dump()
        return response

class Revise(Operator):
    async def __call__(self, problem, solution, feedback, mode: str = None):
        prompt = REVISE_PROMPT.format(problem_description=problem, solution=solution,
        ↪  feedback=feedback)
        fill_kwargs = {"context": prompt, "llm": self.llm}
        if mode:
            fill_kwargs["mode"] = mode
        node = await ActionNode.from_pydantic(ReviseOp).fill(**fill_kwargs)
        response = node.instruct_content.model_dump()
        return response

class Ensemble(Operator):
    async def __call__(self, solutions: List[str], problem: str, mode: str = None):
        answer_mapping = {}
        solution_text = ""
        for index, solution in enumerate(solutions):
            answer_mapping[chr(65 + index)] = index
            solution_text += f"{chr(65 + index)}: \n{str(solution)}\n\n\n"
```

```python
            prompt = ENSEMBLE_PROMPT.format(solutions=solution_text,
            ↪  problem_description=problem)
            fill_kwargs = {"context": prompt, "llm": self.llm}
            if mode:
                fill_kwargs["mode"] = mode
            node = await ActionNode.from_pydantic(EnsembleOp).fill(**fill_kwargs)
            response = node.instruct_content.model_dump()

            answer = response.get("solution_letter", "")
            answer = answer.strip().upper()

            return {"solution": solutions[answer_mapping[answer]]}

class Test(Operator):
    def exec_code(self, solution, entry_point):
        fail_cases = []
        ...
        if fail_cases != []:
            return fail_cases
        else:
            return "no error"

    async def __call__(self, problem, solution, entry_point, test_loop: int = 3):
        for _ in range(test_loop):
            result = self.exec_code(solution, entry_point)
            if result == "no error":
                return {"result": True, "solution": solution}
            elif "exec_fail_case" in result:
                result = result["exec_fail_case"]
                prompt = REFLECTION_ON_PUBLIC_TEST_PROMPT.format(
                    problem=problem,
                    solution=solution,
                    exec_pass=f"executed unsuccessfully, error: \n {result}",
                    test_fail="executed unsucessfully",
                )
                node = await
                ↪  ActionNode.from_pydantic(ReflectionTestOp).fill(context=prompt,
                ↪  llm=self.llm, mode="code_fill")
                response = node.instruct_content.model_dump()
                solution = response["reflection_and_solution"]
            else:
                ...
        result = self.exec_code(solution, entry_point)
        if result == "no error":
            return {"result": True, "solution": solution}
        else:
            return {"result": False, "solution": solution}

class Programmer(Operator):
    async def exec_code(code, timeout=180):
        def run_code():
            try:
                global_namespace = {}

                exec(code, global_namespace)
            except ...

        done_event = threading.Event()
        result = ["Error", "subprocess error"]

        def wrapper():
            nonlocal result
            result = run_code()
            done_event.set()

        with concurrent.futures.ThreadPoolExecutor(max_workers=1) as executor:
            future = executor.submit(wrapper)
            try:
                if done_event.wait(timeout=timeout):
                    return result
                else:
                    future.cancel()
                    return "Error", "Exceed time limit"
            finally:
                executor.shutdown(wait=False)

    async def code_generate(self, problem, analysis, feedback, mode):
        prompt = PYTHON_CODE_VERIFIER_PROMPT.format(problem=problem, analysis=analysis,
        ↪  feedback=feedback)
```

```
        fill_kwargs = {"context": prompt, "llm": self.llm, "function_name": "solve"}
        if mode:
            fill_kwargs["mode"] = mode
        node = await ActionNode.from_pydantic(CodeGenerateOp).fill(**fill_kwargs)
        response = node.instruct_content.model_dump()
        return response

    async def __call__(self, problem: str, analysis: str = "None"):
        code = None
        for i in range(3):
            code = await self.code_generate(problem, analysis, feedback,
            ↪  mode="code_fill")
            code = code["code"]
            status, output = await self.exec_code(code)
            if status == "Success":
                return {"code": code, "output": output}
            else:
                ...
        return {"code": code, "output": "error"}
```

Providing predefined operators can effectively enhance the search efficiency of AFLOW. We implement six common operator structures, including: Generate (Contextual, Code), Format, Review & Revise, Ensemble, Test, and Programmer. For the Test Operator, we use the public test dataset of the dataset as test data. For datasets like MBPP that don't provide a public test dataset, we follow the setting in Zhong et al. (2024a) where we use the first test case of each problem as public test data.

## A.5 MAPPING WORKFLOW FROM FORMULATION TO CODE

**An example of Workflow**

```
async def __call__(self, problem: str, entry_point: str):
    """
    Implementation of the workflow
    Custom operator to generate anything you want.
    But when you want to get standard code, you should use custom_code_generate
    ↪  operator.
    """
    solutions = []
    for _ in range(3):  # Generate 3 solutions
        solution = await self.custom_code_generate(problem=problem,
        ↪  entry_point=entry_point, instruction=prompt_custom.CODE_GENERATE_PROMPT)
        solutions.append(solution['response'])

    best_solution = await self.sc_ensemble(solutions=solutions, problem=problem)

    test_result = await self.test(problem=problem, solution=best_solution['response'],
    ↪  entry_point=entry_point)

    if test_result['result']:
        return test_result['solution'], self.llm.cost_manager.total_cost
    else:
        # If the test fails, try to fix the solution
        fixed_solution = await self.custom(input=f"Problem: {problem}\nFailed solution:
        ↪  {best_solution['response']}\nError: {test_result['solution']}",
        ↪  instruction=prompt_custom.FIX_CODE_PROMPT)
        return fixed_solution['response'], self.llm.cost_manager.total_cost
```

In this example,

- self.custom is the interface for building nodes, through which the Optimizer can generate/modify its prompts.
- self.test and self.sc ensemble are interfaces for using Operators (In this example, this workflow only use 2 operators).
- Edge in AFLOW are represented through code, controlling the flow of all input/output variables between Nodes and Operators to form a complete workflow. Given this definition, the traditional concept of a 'node having two outgoing edges' does not apply to this formulation.

## A.6 MCTS ALGORITHM OF AFLOW.

---

**Algorithm 1** Detailed Explanation of the AFLOW Algorithm

---

**Require:** Initial Workflow $W_0$, Evaluator $G$, Dataset $D$, Number of rounds $N$, Operators $\mathcal{O}$, Top k $k$, Early stopping rounds $n$
**Ensure:** Optimal Workflow $W^*$
 1: Initialize $results \leftarrow \emptyset$, $experiences \leftarrow \emptyset$, $N \leftarrow 20$, $k \leftarrow 3$, $n \leftarrow 5$
 2: $D_V, D_T \leftarrow$ RandomSplit($D$, 0.2, 0.8)        ▷ Split dataset: 20% for validation, 80% for training
 3: $scores \leftarrow$ Execute($W_0, G, D_V$)
 4: $D_V \leftarrow$ SelectHighVarianceInstances($D_V, scores, threshold$)        ▷ Select instances
 5: **for** $round \leftarrow 1$ to $N$ **do**
 6:     **if** $round = 1$ **then**
 7:         $parent \leftarrow W_0$
 8:     **else**
 9:         $parent \leftarrow$ SelectParent($results$)
10:     **end if**
11:     $context \leftarrow$ LoadContext($parent, experiences$)
12:     $W_{round}$,modification$\leftarrow$ Optimizer($context, \mathcal{O}$)
13:     **for** $i \leftarrow 1$ to 5 **do**
14:         $score, cost \leftarrow$ Executor($W_{round}, E, D_V$)
15:         $results$.append($round, score, cost$)
16:     **end for**
17:     $avgScore \leftarrow$ CalculateAverageScore($results[round]$)
18:     $experience \leftarrow$ CreateExperience($parent, modification, avgScore$)
19:     $experiences$.append($experience$)
20:     **if** $avgScore > bestScore$ **then**
21:         $W^* \leftarrow W_{round}$
22:         $bestScore \leftarrow avgScore$
23:     **end if**
24:     **if** The Top $k$ Workflows remains unchanged in $n$ rounds **then**        ▷ Early stopping
           **return** $W^*$
25:     **end if**
26: **end for**
27: **return** $W^*$
28: **procedure** SELECTPARENT($results$)
29:     $sorted\_results \leftarrow$ SortDescending($results$, key=lambda r: r.scores)
30:     $top\_k\_results \leftarrow sorted\_results[: k]$
31:     $scores \leftarrow [result.scores$ for $result$ in $top\_k\_results]$
32:     $probabilities \leftarrow$ CalculateMixedProbabilities($scores$)
33:     **return** SampleFromCategorical(probabilities)
34: **end procedure**
35: **procedure** CALCULATEMIXEDPROBABILITIES($scores$)
36:     $n \leftarrow$ length($scores$), $\lambda \leftarrow 0.4$, $\alpha \leftarrow 0.2$, $s_{max} \leftarrow$ max($scores$)
37:     $w_i \leftarrow \exp(\alpha \cdot (s_i - s_{max}))$ for $i \in [1, n]$
38:     $P_{score} \leftarrow w_i / \sum_{j=1}^{n} w_j$ for $i \in [1, n]$
39:     $P_{uniform} \leftarrow 1/n$ for $i \in [1, n]$
40:     $P_{mixed} \leftarrow \lambda \cdot P_{uniform} + (1 - \lambda) \cdot P_{score}$
41:     **return** $P_{mixed}$
42: **end procedure**
43: **procedure** OPTIMIZER($context, Operators$)
44:     // LLM as Optimizer, generate new workflow and modification.
45:     **return** $newWorkflow, modification$
46: **end procedure**
47: **procedure** EXECUTOR($W, evaluator, dataset$)
48:     // LLM as Executor, execute workflow on dataset and return score and cost
49:     **return** $score, cost$
50: **end procedure**

---

# B  CASE STUDY

## B.1  CASE STUDY OF AFLOW

---

**Alpha Codium like workflow for MBPP**

---

```python
CODE_GENERATE_PROMPT = """
Generate a Python function to solve the given problem. Ensure the function name matches
↪  the one specified in the problem. Include necessary imports. Use clear variable
↪  names and add comments for clarity.

Problem:
{problem}

Function signature:
{entry_point}

Generate the complete function below:
"""

FIX_CODE_PROMPT = """
The provided solution failed to pass the tests. Please analyze the error and fix the
↪  code. Ensure the function name and signature remain unchanged. If necessary, add or
↪  modify imports, correct logical errors, and improve the implementation.

Problem:
{input}

Provide the corrected function below:
"""

GENERATE_TESTS_PROMPT = """
Given the problem and a potential solution, generate additional test cases to
↪  thoroughly evaluate the function. Include edge cases and typical scenarios. Format
↪  the test cases as assert statements that can be directly added to a Python test
↪  function.

Problem:
{input}

Generate 3-5 additional test cases as assert statements:
"""

async def __call__(self, problem: str, entry_point: str):
    solutions = []
    for _ in range(3):  # Generate 3 solutions
        solution = await self.custom_code_generate(problem=problem,
        ↪  entry_point=entry_point, instruction=prompt_custom.CODE_GENERATE_PROMPT)
        solutions.append(solution['response'])
    best_solution = await self.sc_ensemble(solutions=solutions, problem=problem)
    # Generate additional test cases
    additional_tests = await self.custom(input=f"Problem: {problem}\nSolution:
    ↪  {best_solution['response']}", instruction=prompt_custom.GENERATE_TESTS_PROMPT)
    # Combine original problem and additional tests
    enhanced_problem = f"{problem}\n\nAdditional test
    ↪  cases:\n{additional_tests['response']}"
    test_result = await self.test(problem=enhanced_problem,
    ↪  solution=best_solution['response'], entry_point=entry_point)
    if test_result['result']:
        return test_result['solution'], self.llm.cost_manager.total_cost
    else:
        # If the test fails, try to fix the solution
        fixed_solution = await self.custom(input=f"Problem: {problem}\nFailed solution:
        ↪  {best_solution['response']}\nError: {test_result['solution']}",
        ↪  instruction=prompt_custom.FIX_CODE_PROMPT)
        return fixed_solution['response'], self.llm.cost_manager.total_cost
```

---

AFLOW demonstrates its ability to reduce human effort by evolving from an empty workflow to a solution highly similar to manually designed workflows like Ridnik et al. (2024) in the code generation scenario. This showcases AFLOW's capability to generate efficient workflows comparable to expert designs with minimal human intervention.

**The optimal workflow generated for MATH**

```
REFINE_ANSWER_PROMPT = """
Given the mathematical problem and the output from the code execution, please provide a
↪   well-formatted and detailed solution. Follow these guidelines:
1. Begin with a clear statement of the problem.
2. Explain the approach and any formulas or concepts used.
3. Show step-by-step calculations, using LaTeX notation for mathematical expressions.
4. Interpret the code output and incorporate it into your explanation.
5. Provide a final answer, enclosed in \boxed{} LaTeX notation.
6. Ensure all mathematical notation is in LaTeX format.
Your response should be comprehensive, mathematically rigorous, and easy to follow.
"""
GENERATE_SOLUTION_PROMPT = """
Please solve the given mathematical problem step by step. Follow these guidelines:
1. State the problem clearly.
2. Outline the approach and any relevant formulas or concepts.
3. Provide detailed calculations, using LaTeX notation for mathematical expressions.
4. Explain each step of your reasoning.
5. Present the final answer enclosed in \boxed{} LaTeX notation.
6. Ensure all mathematical notation is in LaTeX format.
Your solution should be thorough, mathematically sound, and easy to understand.
"""
DETAILED_SOLUTION_PROMPT = """
Provide a comprehensive, step-by-step solution to the given mathematical problem. Your
↪   response should include:
1. A clear restatement of the problem.
2. An explanation of the mathematical concepts and theorems involved.
3. A detailed, logical progression of steps leading to the solution.
4. Clear explanations for each step, including the reasoning behind it.
5. All mathematical expressions and equations in LaTeX format.
6. Visual aids or diagrams if applicable (described in text).
7. A final answer clearly marked and enclosed in \boxed{} LaTeX notation.
8. A brief explanation of the significance of the result, if relevant.
Ensure your solution is rigorous, easy to follow, and educational for someone learning
↪   the concept.
"""
async def __call__(self, problem: str):
    """
    Implementation of the graph
    """
    # Use Programmer to generate and execute Python code
    code_solution = await self.programmer(problem=problem)
    # Use Custom to refine and format the answer
    refined_solution = await self.custom(input=problem + f"\nCode output:
    ↪   {code_solution['output']}", instruction=prompt_custom.REFINE_ANSWER_PROMPT)
    # Generate a detailed step-by-step solution using Custom
    detailed_solution = await self.custom(input=problem,
    ↪   instruction=prompt_custom.DETAILED_SOLUTION_PROMPT)
    # Generate multiple solutions using Custom
    solutions = [
        refined_solution['response'],
        detailed_solution['response']
    ]
    for _ in range(2):
        solution = await self.custom(input=problem,
        ↪   instruction=prompt_custom.GENERATE_SOLUTION_PROMPT)
        solutions.append(solution['response'])
    # Use ScEnsemble to select the best solution
    final_solution = await self.sc_ensemble(solutions=solutions, problem=problem)
    return final_solution['response'], self.llm.cost_manager.total_cost
```

This optimal workflow generated for the MATH task showcases the model's ability to generate complex, task-specific solutions from task-agnostic initial settings. It combines programmatic solutions with various reasoning strategies, culminating in an ensemble selection process, and spontaneously formats the answer into the required form. This adaptation demonstrates the model's flexibility in tailoring workflows to different problem domains, while maintaining sophisticated problem-solving structures.

**The optimal workflow generated for MBPP**

```
CODE_GENERATE_PROMPT = """
Generate a Python function to solve the given problem. Ensure the function name matches
↪   the one specified in the problem. Include necessary imports. Use clear variable
↪   names and add comments for clarity.

Problem:
{problem}

Function signature:
{entry_point}

Generate the complete function below:
"""

FIX_CODE_PROMPT = """
The provided solution failed to pass the tests. Please analyze the error and fix the
↪   code. Ensure the function name and signature remain unchanged. If necessary, add or
↪   modify imports, correct logical errors, and improve the implementation.

Problem:
{input}

Provide the corrected function below:
"""

async def __call__(self, problem: str, entry_point: str):
    """
    Implementation of the workflow
    Custom operator to generate anything you want.
    But when you want to get standard code, you should use custom_code_generate
    ↪   operator.
    """
    solutions = []
    for _ in range(3):  # Generate 3 solutions
        solution = await self.custom_code_generate(problem=problem,
        ↪   entry_point=entry_point, instruction=prompt_custom.CODE_GENERATE_PROMPT)
        solutions.append(solution['response'])

    best_solution = await self.sc_ensemble(solutions=solutions, problem=problem)

    test_result = await self.test(problem=problem, solution=best_solution['response'],
    ↪   entry_point=entry_point)

    if test_result['result']:
        return test_result['solution'], self.llm.cost_manager.total_cost
    else:
        # If the test fails, try to fix the solution
        fixed_solution = await self.custom(input=f"Problem: {problem}\nFailed solution:
        ↪   {best_solution['response']}\nError: {test_result['solution']}",
        ↪   instruction=prompt_custom.FIX_CODE_PROMPT)
        return fixed_solution['response'], self.llm.cost_manager.total_cost
```

The optimal workflow generated for the MBPP task simply combines operators with an ingenious FIX-CODE PROMPT, achieving the optimal workflow in the iteration at the fourteenth round. Although this workflow is simple, its score is extremely high and stable, demonstrating AFLOW's potential to find the optimal cost-performance balance.

**The optimal workflow generated for HotpotQA**

```
FORMAT_ANSWER_PROMPT = """
Given the question and the best answer, format the final answer to be concise,
↪   accurate, and directly addressing the question. Ensure the answer is a clear, brief
↪   statement without additional explanation or reasoning. If the answer is a name,
↪   profession, or short phrase, provide only that information without forming a
↪   complete sentence.

For example:
- If the answer is a person's name, just provide the name.
- If the answer is a profession, state only the profession.
- If the answer is a short phrase, give only that phrase.
```

```
Do not include any prefixes like "The answer is" or "The profession is". Just provide
↪  the answer itself.
"""

async def __call__(self, problem: str):
    """
    Implementation of the workflow
    """
    solutions = []
    for _ in range(3):
        initial_response = await self.answer_generate(input=problem)
        thought_process = initial_response['thought']
        initial_answer = initial_response['answer']
        solutions.append(initial_answer)

    ensemble_result = await self.sc_ensemble(solutions=solutions)
    best_answer = ensemble_result['response']

    refined_solution = await self.custom(
        input=f"Question: {problem}\nBest answer: {best_answer}",
        instruction=prompt_custom.FORMAT_ANSWER_PROMPT
    )

    return refined_solution['response'], self.llm.cost_manager.total_cost
```

The optimal workflow generated for the HotpotQA task demonstrates the effectiveness of execution feedback. Apart from logical reasoning, another factor affecting QA problem scores is effective formatting. AFLOW can effectively identify the correct format and automatically perform formatting through learning from execution feedback, showcasing the efficacy of this design.

**An ensemble structure that emerged in the GSM8K ablation experiment**

```
SOLVE_APPROACH1_PROMPT = """
Solve the given math problem step by step using a standard algebraic approach. After
↪  solving, extract the final numerical answer and format it as follows:

Final Answer: [Insert the numerical value here]

Ensure that only the numerical value is provided after "Final Answer:", without any
↪  units or additional text.

Problem:
"""

SOLVE_APPROACH2_PROMPT = """
Solve the given math problem step by step using a visual or diagrammatic approach, if
↪  applicable. If not applicable, use an alternative method different from the
↪  standard algebraic approach. After solving, extract the final numerical answer and
↪  format it as follows:

Final Answer: [Insert the numerical value here]

Ensure that only the numerical value is provided after "Final Answer:", without any
↪  units or additional text.

Problem:
"""

SOLVE_APPROACH3_PROMPT = """
Solve the given math problem step by step using estimation or approximation techniques,
↪  then refine the answer for accuracy. After solving, extract the final numerical
↪  answer and format it as follows:

Final Answer: [Insert the numerical value here]

Ensure that only the numerical value is provided after "Final Answer:", without any
↪  units or additional text.

Problem:
"""

COMPARE_AND_SELECT_PROMPT = """
```

```
Compare the three solutions provided for the given math problem. Analyze each solution
↪   for correctness, completeness, and consistency with the problem statement. Select
↪   the most accurate and reliable solution, or if all solutions agree, confirm their
↪   consistency.

If the solutions differ, explain the differences and justify your selection of the most
↪   accurate answer. If all solutions agree, state this consistency.

Provide the final answer in the following format:

Final Answer: [Insert the numerical value here]

Ensure that only the numerical value is provided after "Final Answer:", without any
↪   units or additional text.

Problem:
"""

async def __call__(self, problem: str):
    """
    Implementation of the workflow
    """
    solution1 = await self.custom(input=problem,
    ↪   instruction=prompt_custom.SOLVE_APPROACH1_PROMPT)
    solution2 = await self.custom(input=problem,
    ↪   instruction=prompt_custom.SOLVE_APPROACH2_PROMPT)
    solution3 = await self.custom(input=problem,
    ↪   instruction=prompt_custom.SOLVE_APPROACH3_PROMPT)
    combined_solutions = f"Solution 1: {solution1['response']}\nSolution 2: 
    ↪   {solution2['response']}\nSolution 3: {solution3['response']}"
    final_solution = await self.custom(input=problem + "\n" + combined_solutions,
    ↪   instruction=prompt_custom.COMPARE_AND_SELECT_PROMPT)
    return final_solution['response'], self.llm.cost_manager.total_cost
```

In the ablation study, where predefined operators were deliberately removed, AFLOW surprisingly developed this simplified yet effective workflow. Most notably, it independently evolved an ensemble-like operator, mirroring a key aspect of the optimal workflow. This emergence of a multi-solution generation and selection process, despite reduced guidance, highlights AFLOW's inherent tendency towards robust problem-solving strategies. The spontaneous development of this ensemble approach in a constrained environment underscores AFLOW's ability to identify and implement effective techniques, even when operating with limited resources or instructions. This unexpected convergence between the ablated and optimal workflows further demonstrates AFLOW's capacity for developing sophisticated, human-like problem-solving paradigms across different experimental conditions.

## B.2 CASE STUDY OF ADAS

**Iterative Knowledge-Enhanced Refinement workflow for HotpotQA**

```
async def forward(self, taskInfo):
    import asyncio

    # Step 1: Initial reasoning by diverse expert agents
    initial_instruction = 'Please think step by step and solve the task based on your
    ↪   expertise.'
    expert_agents = [
        LLMAgentBase(['thinking', 'answer'], 'Expert Agent', role=role,
        ↪   temperature=0.7)
        for role in ['Reading Specialist', 'Logic Specialist', 'Generalist']
    ]

    async def run_expert(agent):
        return await agent([taskInfo], initial_instruction)

    initial_results = await asyncio.gather(*[run_expert(agent) for agent in
    ↪   expert_agents])

    combined_infos = [taskInfo] + [info for result in initial_results for info in
    ↪   result]  # Flattening initial_results
```

```python
        # Step 2: Iterative refinement with external knowledge integration
    max_iterations = 2
    for iteration in range(max_iterations):
        # Retrieve external knowledge
        knowledge_retrieval_instruction = 'Retrieve relevant information from a
        ↪   knowledge base that can assist in refining the solution.'
        knowledge_retrieval_agent = LLMAgentBase(['retrieved_info'], 'Knowledge
        ↪   Retrieval Agent')
        retrieved_results = await knowledge_retrieval_agent(combined_infos,
        ↪   knowledge_retrieval_instruction)
        retrieved_info = retrieved_results[0]

        # Verify external knowledge
        verification_instruction = 'Verify the relevancy and accuracy of the retrieved
        ↪   information.'
        verification_agent = LLMAgentBase(['verified_info'], 'Verification Agent')
        verified_results = await verification_agent([taskInfo, retrieved_info],
        ↪   verification_instruction)
        verified_info = verified_results[0]

        # Refinement phase using verified knowledge
        refinement_instruction = 'Review and refine the insights provided by other
        ↪   agents using the verified external knowledge.'
        refinement_agents = [
            LLMAgentBase(['refined_thinking', 'refined_answer'], 'Refinement Agent',
            ↪   role=role, temperature=0.5)
            for role in ['Reading Specialist', 'Logic Specialist', 'Generalist']
        ]
        combined_infos_with_verification = combined_infos + [verified_info]

        async def run_refinement(agent):
            return await agent(combined_infos_with_verification,
            ↪   refinement_instruction)

        refinement_results = await asyncio.gather(*[run_refinement(agent) for agent in
        ↪   refinement_agents])
        combined_infos.extend([info for result in refinement_results for info in
        ↪   result])  # Flattening refinement_results

    # Step 3: Final synthesis agent integrates all refined insights
    final_decision_instruction = 'Synthesize all refined insights and provide a final
    ↪   answer.'
    final_decision_agent = LLMAgentBase(['thinking', 'answer'], 'Final Decision Agent',
    ↪   temperature=0.3)
    final_thinking, final_answer = await final_decision_agent(combined_infos,
    ↪   final_decision_instruction)

    return final_answer}
```

When designing workflows, ADAS incorporates all workflows from the search history into the prompt, distinguishing them only by their generation order and scores. However, the complex information embedded in the intricate structure of workflows, coupled with the accumulation of search iterations, the vast amount of information, and the continuously accumulating irrelevant information, poses significant challenges for LLM reasoning. ADAS stores experience from previous searches at the coarsest granularity—directly storing all complete workflows. This approach causes the LLM designing workflows in ADAS to behave more like an explorer of infinite possibilities within $\mathcal{E}$ rather than a designer seeking the optimal workflow.

As shown in the code in Appendix B.2, the optimal workflow discovered by ADAS assigns diverse roles and multiple steps for refinement and summarization. However, for multi-hop reasoning tasks, the correct approach is to continuously reduce the problem scale to single-hop reasoning. Contrary to this, ADAS's optimal workflow actually increases the problem scale, ultimately attempting to use the LLM's summarization ability to synthesize information, rather than gradually reducing the number of hops based on the characteristics of multi-hop reasoning scenarios.

## C  OPTIMIZATION PROCESS OF AFLOW

Taking AFLOW's search process on the Math dataset as an example, we demonstrate how AFLOW iteratively improves workflows based on tree-structured experience and execution feedback.

## C.1 TREE-STRUCTURED EXPERIENCE.

**Processed Experience (formatted as tree sturcture)**

```
{
    "1": {
        "score": 0.4873949579831933,
        "success": {
            "2": {
                "modification": "Add the Programmer operator to generate and execute
                ↪ Python code for mathematical calculations, and use the Custom
                ↪ operator to refine and format the final answer.",
                "score": 0.5243697478991597
            }
        },
        "failure": {
            "8": {
                "modification": "Add a ScEnsemble operator to generate multiple
                ↪ solutions and select the best one. This will help improve the
                ↪ accuracy of the final answer.",
                "score": 0.4336134453781512
            }
        }
    },
    "2": {
        "score": 0.5243697478991597,
        "success": {
            "3": {
                "modification": "Add a ScEnsemble operator to improve the reliability
                ↪ of the final answer by generating multiple solutions and selecting
                ↪ the most consistent one.",
                "score": 0.5277310924369747
            }
        },
        "failure": {
            "6": {
                "modification": "Add a ScEnsemble operator to improve the reliability
                ↪ of the final answer by generating multiple solutions and selecting
                ↪ the most consistent one.",
                "score": 0.4722689075630252
            },
            "7": {
                "modification": "Add a ScEnsemble operator to improve the reliability
                ↪ of the final answer by generating multiple solutions and selecting
                ↪ the most consistent one.",
                "score": 0.5243697478991597
            }
        }
    },
    "3": {
        "score": 0.5277310924369748,
        "success": {
            "14": {
                "modification": "Modify the Custom operator to generate a more detailed
                ↪ step-by-step solution, and add a new Custom operator to review and
                ↪ refine the final answer. This will improve the clarity, accuracy,
                ↪ and completeness of the solution process.",
                "score": 0.5310924369747899
            },
            "5": {
                "modification": "Add a new Custom operator to generate a detailed
                ↪ step-by-step solution, and modify the ScEnsemble operator to
                ↪ compare and select the best solution from multiple approaches.",
                "score": 0.5512605042016807
            },
            "9": {
                "modification": "Add a new Custom operator to generate a detailed
                ↪ step-by-step solution, and modify the ScEnsemble operator to
                ↪ compare and select the best solution from multiple approaches.",
                "score": 0.5378151260504201
            }
        },
        "failure": {
            "10": {
                "modification": "Add a new Custom operator to generate a step-by-step
                ↪ solution, and modify the ScEnsemble operator to compare and select
                ↪ the best solution from multiple approaches.",
```

```
                "score": 0.5042016806722688
            },
            "13": {
                "modification": "Modify the Custom operator to generate a more detailed
                ↪   step-by-step solution, and add a new Custom operator to refine and
                ↪   format the final answer. This will improve the clarity and accuracy
                ↪   of the solution process.",
                "score": 0.5193277310924369
            },
            "4": {
                "modification": "Add a new Custom operator to generate multiple
                ↪   solutions using different approaches, then use ScEnsemble to select
                ↪   the best solution. This will increase the diversity of solutions
                ↪   and potentially improve accuracy.",
                "score": 0.0
            }
        }
    },
    "9": {
        "score": 0.5378151260504203,
        "success": {},
        "failure": {
            "11": {
                "modification": "Add a new Custom operator to generate a detailed
                ↪   step-by-step solution with explanations, and incorporate it into
                ↪   the ensemble process. This will provide a more comprehensive
                ↪   approach to solving math problems.",
                "score": 0.5159663865546219
            },
            "12": {
                "modification": "Add a new Custom operator to generate multiple
                ↪   solution approaches, then use ScEnsemble to select the best
                ↪   solution. This will increase the diversity of solutions and
                ↪   potentially improve accuracy.",
                "score": 0.0
            },
            "16": {
                "modification": "Add a new Custom operator to generate multiple
                ↪   solution approaches, then use ScEnsemble to select the best
                ↪   solution. This will increase the diversity of solutions and
                ↪   potentially improve accuracy.",
                "score": 0.5210084033613446
            }
        }
    },
    "14": {
        "score": 0.5310924369747899,
        "success": {},
        "failure": {
            "15": {
                "modification": "Add a new Custom operator to generate multiple
                ↪   solutions, then use ScEnsemble to select the best one. This
                ↪   modification aims to improve the accuracy and consistency of the
                ↪   final answer.",
                "score": 0.5243697478991596
            },
            "18": {
                "modification": "Add a new Custom operator to generate a more detailed
                ↪   step-by-step solution, and modify the ScEnsemble operator to
                ↪   compare and select the best solution from multiple generated
                ↪   solutions.",
                "score": 0.5176470588235293
            }
        }
    },
    "5": {
        "score": 0.5512605042016807,
        "success": {},
        "failure": {
            "17": {
                "modification": "Add a new Custom operator to generate multiple
                ↪   solutions using different approaches, and modify the ScEnsemble
                ↪   operator to select the best solution from a larger pool of
                ↪   candidates.",
                "score": 0.0
            },
            "19": {
```

```
                "modification": "Add a new Custom operator to generate a simplified
                ↪   solution, which will be used alongside the existing detailed
                ↪   solution to provide a more comprehensive answer. This simplified
                ↪   solution will be added to the list of solutions for the ScEnsemble
                ↪   operator to consider.",
                "score": 0.5445378151260505
            }
        }
    }
}
```

More optimization trajectories will be made available in an open-source repository upon publication.

**Less Effective Optimization Steps:**

- Round 1 → Round 8 (Score decreased from 0.4873 to 0.4336): The key change was the removal of the Programmer operator, relying solely on Custom + ScEnsemble, which lost the computational precision provided by programmatic solutions, demonstrating that removing concrete computational capabilities significantly hurts performance.

- Round 9 → Round 16 (Score decreased from 0.5378 to 0.5210): The key change involved simplifying the solution generation process without maintaining the review step, which resulted in the loss of the quality control aspect of solution refinement, showing that solution quality checks are important for maintaining performance.

**Successful Optimization Steps:**

- Round 1 → Round 2 (Score improved from 0.4874 to 0.5244): The addition of the Programmer operator to generate executable Python code and the use of the Custom operator to refine results introduced concrete computational capabilities alongside human-like reasoning, creating a more robust solution approach and providing a foundation for both numerical accuracy and explanation quality.

- Round 2 → Round 3 (Score improved from 0.5244 to 0.5277): The introduction of the ScEnsemble operator to select from multiple solutions added solution diversity and reliability through ensemble selection, creating a more robust system by considering multiple solution approaches.

- Round 3 → Round 5 (Score improved from 0.52773 to 0.5513): The key change was the addition of detailed step-by-step solution generation, which enhanced solution clarity and comprehensiveness, ultimately improving the pedagogical value of solutions while maintaining accuracy.

**The tree-structured optimization process helped guide LLM workflow improvements in several ways:**

- **Path Discovery**: The tree structure allowed exploration of multiple optimization directions simultaneously, enabling the development of successful paths while pruning less successful branches, which facilitated the efficient discovery of effective combinations of operators.

- **Incremental Improvement**: Each node in the tree represents a specific workflow configuration, and the success/failure feedback at each step helped identify which modifications were beneficial, with the scoring system providing quantitative guidance for optimization decisions.

- **Pattern Recognition**: The tree structure made it easier to identify patterns in successful versus unsuccessful modifications, revealing common elements in high-scoring branches, such as the combination of Programmer + Custom + ScEnsemble, which informed future optimization decisions.

- **Error Recovery**: When a modification led to decreased performance, the tree structure facilitated easy backtracking, allowing exploration of alternative optimization paths from previous successful states and preventing the process from getting stuck in local optima.

## C.2 EXECUTION FEEDBACK.

In the process of optimizing the overall response generation, we implemented a concise and task-agnostic prompt: "Below are the logs of some results with the aforementioned Graph that performed well but encountered errors, which can be used as references for optimization: {log}". This approach enabled execution feedback to assist LLM in workflow optimization. The following examples demonstrate how adjustments to the prompts enhanced the quality, consistency, and clarity of the answers, with particular emphasis on answer formatting as a key illustration.

**Optimization Steps:**

- Round 1 → Round 2: REFINE_ANSWER_PROMPT adds "Provide a final answer, enclosed in \boxed{} LaTeX notation", resulting in the ability to identify patterns in the scoring feedback without knowing the specific rules of the scoring function.

- Round 1 → Round 8: While REFINE_ANSWER_PROMPT adds "Provide a clear, concise final answer" to shift focus towards presenting answers more concisely, Round 8 lacks the strict LaTeX formatting constraints present in Round 2. This leads to Round 8 scoring lower than the baseline Round 1. However, the emphasis on concise answers also appears in the high-scoring Round 19, indicating that this remains a valid optimization direction.

- Round 3 → Round 13: REFINE_ANSWER_PROMPT introduces two new requirements: "If there are multiple possible answers, list all of them separated by commas within the \boxed{}" and "Simplify expressions where possible without losing accuracy". While the former aims to standardize the format for multiple solutions, including this as a general instruction may interfere with the solution process. Statistical analysis shows that Round 13 contains 55 instances of comma-separated answers, notably higher than better-performing rounds such as Round 5 (43 instances) and Round 9 (42 instances).

## D  PARETO FRONT: DETAILED COST-PERFORMANCE DATA

Detailed cost-performance data for HumanEval. Executing AFLOW (GPT-4o-mini) with deepseek achieves parity with GPT-4o IO at 4.55% of the cost. Executing AFLOW (deepseek) with deepseek and AFlow(GPT-4o-mini) with GPT-4o-mini outperform GPT-4o IO at 5.92% and 8.05% of the cost, respectively.

| Model | Method | Score (%) | Cost ($) |
|---|---|---|---|
| gpt-4o-mini | IO | 0.8702 | 0.0223 |
| gpt-4o-mini | CoT | 0.8860 | 0.0277 |
| gpt-4o-mini | CoT SC | 0.9160 | 0.1794 |
| gpt-4o-mini | MedPrompt | 0.9160 | 0.2200 |
| gpt-4o-mini | LLM Debate | 0.8930 | 0.2278 |
| gpt-4o-mini | Self Refine | 0.8780 | 0.1232 |
| gpt-4o-mini | AFLOW (gpt-4o-mini) | 0.9470 | 0.0513 |
| gpt-4o-mini | AFLOW (deepseek) | 0.9084 | 0.0669 |
| deepseek | IO | 0.8860 | 0.0127 |
| deepseek | CoT | 0.8930 | 0.0180 |
| deepseek | CoT SC | 0.8860 | 0.1168 |
| deepseek | MedPrompt | 0.8860 | 0.1433 |
| deepseek | LLM Debate | 0.8930 | 0.1484 |
| deepseek | Self Refine | 0.9000 | 0.0802 |
| deepseek | AFLOW (gpt-4o-mini) | 0.9390 | 0.0291 |
| deepseek | AFLOW (deepseek) | 0.9466 | 0.0377 |
| gpt-4o | IO | 0.9389 | 0.6371 |
| gpt-4o | CoT | 0.9310 | 0.7772 |
| gpt-4o | CoT SC | 0.9470 | 5.0345 |
| gpt-4o | MedPrompt | 0.9390 | 6.1756 |
| gpt-4o | LLM Debate | 0.9470 | 6.3952 |
| gpt-4o | Self Refine | 0.9160 | 3.4589 |
| gpt-4o | AFLOW (gpt-4o-mini) | 0.9620 | 1.0111 |
| gpt-4o | AFLOW (deepseek) | 0.9542 | 1.6600 |
| claude-3.5-sonnet | IO | 0.9084 | 0.6987 |
| claude-3.5-sonnet | CoT | 0.9240 | 0.6412 |
| claude-3.5-sonnet | CoT SC | 0.9390 | 4.1534 |
| claude-3.5-sonnet | MedPrompt | 0.9160 | 5.0949 |
| claude-3.5-sonnet | LLM Debate | 0.9080 | 5.2761 |
| claude-3.5-sonnet | Self Refine | 0.8930 | 2.8536 |
| claude-3.5-sonnet | AFLOW (gpt-4o-mini) | 0.9540 | 1.1612 |
| claude-3.5-sonnet | AFLOW (deepseek) | 0.9466 | 1.3252 |

## E  DISCUSSION ON WORKFLOW, AGENTIC WORKFLOW, MULTIAGENT SYSTEMS

In the field of LLM applications, concepts such as Workflow, Agentic Workflow, and MultiAgent Systems are frequently referenced, each with distinct emphases. The notion of Workflow in (Qiao et al., 2025) primarily focuses on the structured decomposition of real-world tasks, such as generating execution sequences for specific tasks in digital environments. (Qiao et al., 2025) work represents a significant advancement in workflow benchmarking through its comprehensive multi-faceted approach spanning function calls, problem-solving, embodied planning, and open-grounded planning scenarios.

Despite terminological differences, the multi-agent systems described in Zhang et al. (2024a;b); Zhuge et al. (2024) are functionally equivalent to agentic workflows. Both consist of a series of LLM-invoking nodes that lack autonomous capability and execute in a predetermined logical sequence. Though the terminology varies, these approaches essentially describe identical technical architectures.

Meanwhile, Niu et al. (2025) explores methods for dynamically adjusting workflows during task execution, enhancing the system's ability to address complex problems by modifying workflows in real-time based on execution feedback.

AFLOW proposed in this paper focuses on the automated generation and optimization of workflows, aiming to reduce dependence on manually designed initial workflows while improving the scalability and generalizability of agentic workflows. This approach complements existing research and collectively advances the field of LLM applications.

# F DISCUSSION ON OPEN-ENDED TASKS

## F.1 ADAPTING AFLOW FOR OPEN-ENDED TASKS

While we emphasized AFLOW's powerful capabilities in solving reasoning tasks with numerical feedback in the main body, there are many tasks in the real world without numerical feedback and non-reasoning requirements (Dai et al., 2024). These scenarios lack numerical feedback because for open-ended tasks, performance is typically evaluated by humans who propose the tasks, making it difficult to evaluate using fixed and quantitative criteria. By introducing LLM as a judge (Zheng et al., 2023), this challenge can be addressed to some extent, as the LLM-as-judge evaluation prompt can effectively incorporate human preferences to adaptively evaluate open-ended tasks. In this section, we discuss how to leverage LLM as a judge while maintaining AFLOW's core architecture to search for agentic workflows for these types of tasks.

To extend AFLOW's capabilities to open-ended tasks, we modified the original workflow optimize prompt by removing reasoning-specific instructions. The new prompt is as follows:

---

**Workflow optimize prompt for open-ended tasks**

---

```
PROMPT = """You are constructing a graph and corresponding prompts to jointly solve
↪  {type} problems. Referring to the provided graph and prompts, which form a basic
↪  example of a {type} solution approach, please reconstruct and optimize them. You
↪  may add, modify, or delete nodes, parameters, or prompts. Include your single
↪  modification enclosed in XML tags in your reply. Ensure they are complete and
↪  correct to avoid runtime failures.Use logical and control flow (such as IF-ELSE
↪  and loops) to achieve more advanced code representation.Ensure all prompts
↪  required by the current graph are included in `prompt_custom`. Do not include any
↪  additional prompts. The prompts you need to generate are limited to those used in
↪  `prompt_custom.XXX`. Other methods already have built-in prompts and are
↪  prohibited from being generated. Generate only the prompts needed by the graph and
↪  remove any unused prompts from `prompt_custom`.The generated prompts must not
↪  contain any placeholders. Considering information loss, complex graphs may yield
↪  better results, but insufficient information transmission might omit the solution.
↪  Ensure that the necessary context is included throughout the process.
"""
```

---

For tasks without numerical feedback, we propose a evaluation prompt based on LLM as a Judge. We utilize GPT-4o as our evaluation model, implementing the following prompt for scoring open-ended tasks:

---

**Evaluation prompt**

---

```
EVAL_PROMPT = """You are an expert evaluator tasked with scoring responses to
↪   open-ended questions. You will be provided with:
1. The original question/prompt
2. A golden (reference) answer (if available)
```

---

```
3. A candidate response to be evaluated

Please evaluate the candidate response on the following dimensions, each scored from
↪   1-5. When no reference answer is provided, use your expert judgment to assess the
↪   expected quality level for the given task type:

1. Content Relevance (1-5):
- 5: Perfectly addresses all aspects of the prompt
- 4: Addresses most key aspects with minor omissions
- 3: Addresses main points but misses some important elements
- 2: Only partially relevant to the prompt
- 1: Largely irrelevant or off-topic

2. Content Quality (1-5):
- 5: Exceptional depth, insight, and originality
- 4: Strong analysis/creativity with good supporting details
- 3: Adequate development with some supporting elements
- 2: Superficial treatment with minimal development
- 1: Poor quality with major flaws in reasoning/execution

3. Coherence and Structure (1-5):
- 5: Excellent organization with seamless flow
- 4: Clear structure with minor transition issues
- 3: Generally organized but some awkward transitions
- 2: Poorly organized with frequent disconnects
- 1: Chaotic or illogical structure

4. Reference Comparison (1-5):
- 5: Matches or exceeds expected quality for this type of task
- 4: Slightly below ideal but strong performance
- 3: Moderately below ideal but acceptable
- 2: Significantly below expected quality
- 1: Far below acceptable quality standards

Please provide:
1. Numeric scores for each dimension (1-5)
2. Brief justification for each score (1-2 sentences)
3. Total score (sum of the four dimensions, maximum 20 points)
4. Summary feedback (2-3 sentences)

Format your response as:
Content Relevance: [score] points
- Justification: [brief explanation]

Content Quality: [score] points
- Justification: [brief explanation]

Coherence: [score] points
- Justification: [brief explanation]

Reference Comparison: [score] points
- Justification: [brief explanation]

Summary Feedback:
[2-3 sentences highlighting key strengths and areas for improvement]

<score>[sum of all dimensions](pure number)</score>

"""
```

This evaluation framework replaces the original evaluation function in AFLOW's executing evaluation stage, enabling assessment of open-ended tasks while maintaining consistent evaluation standards.

## F.2 CASE STUDY

We demonstrate the effectiveness of our adapted AFLOW through two open-ended task scenarios: long-form novel generation and academic idea generation. Both cases lack numerical feedback and utilize our adapted methodology. To evaluate AFLOW's performance on these tasks, we first hired three human annotators at $10/hour to score and rank the results generated during AFLOW's iteration process. Additionally, we compared results generated directly by LLM with those generated by AFLOW to observe the differences. Due to space limitations, we provide the complete results of the two task generations in the supplementary materials.

### F.2.1 LONG-FORM NOVEL GENERATION

In this case study, we employed Claude-3.5-sonnet as the execution and optimization model and GPT-4o as the evaluation model due to its strong language capabilities. The system was tested with a single question without reference answers:

---

**Novel data**

---

```
{
"question":"Create a novel-length narrative, exactly 20,000 words long (please ensure
↪  this specific length is met), exploring a world where time's flow varies for each
↪  person based on their deepest regrets. The story should meticulously examine how
↪  emotional burdens affect temporal perception, with careful attention to
↪  maintaining the required length. Focus on developing several characters whose
↪  unique regrets create distinctly different experiences of time's passage.",
"requirement":"Write a long novel with emphasis on substantial length, logical
↪  interconnections between chapters, and refined language style.",
"answer": "No reference answer"
}
```

---

The optimized workflow obtained after eight iterations of AFLOW was:

---

**Novel generation workflow**

---

```
OUTLINE_PROMPT = """
Create a detailed outline for a novel based on the given requirements. Include:
1. A list of main characters with their deepest regrets and how it affects their
↪  perception of time
2. A chapter-by-chapter breakdown of the plot, ensuring logical interconnections
3. Key themes and motifs to be explored throughout the novel
4. A rough word count estimate for each chapter to aim for the required total length

Provide this outline in a structured format.
"""

CHARACTER_PROFILE_PROMPT = """
Based on the given requirements and outline, create detailed character profiles for
↪  each main character. For each character, include:
1. Name, age, and physical description
2. Background and personal history
3. Their deepest regret and how it affects their perception of time
4. Personality traits, motivations, and goals
5. Relationships with other characters
6. Character arc throughout the novel

Provide these profiles in a structured format.
"""

CHAPTER_PROMPT = """
Write a single chapter of the novel based on the given requirements, provided outline,
↪  and character profiles. Follow these guidelines:
1. Adhere to the chapter structure from the outline
2. Maintain logical interconnections with previous and future chapters
3. Use refined language and vivid descriptions
4. Develop characters' arcs based on their regrets and time perception
5. Incorporate the key themes and motifs
6. Aim for approximately 10,000 words per chapter

Write the complete chapter without stopping or summarizing. Do not include any
↪  meta-commentary or explanations outside the chapter text itself.
"""

    async def __call__(self, problem: str):
        """
        Implementation of the workflow
        """
        outline = await self.custom(input=problem,
        ↪  instruction=prompt_custom.OUTLINE_PROMPT)
        character_profiles = await self.custom(input=problem + f"\nOutline:
        ↪  {outline['response']}",
        ↪  instruction=prompt_custom.CHARACTER_PROFILE_PROMPT)
```

---

```
        novel = ""
        chapter_count = 10   # Adjust as needed
        for i in range(chapter_count):
            chapter = await self.custom(
                input=problem + f"\nOutline: {outline['response']}\nCharacter
                ↪   Profiles: {character_profiles['response']}\nChapter Number:
                ↪   {i+1}",
                instruction=prompt_custom.CHAPTER_PROMPT
            )
            novel += f"\n\nChapter {i+1}\n\n{chapter['response']}"

        return novel, self.llm.cost_manager.total_cost
```

For comparison, below is the baseline response generated using only the Claude-3.5-sonnet, without any AFLOW optimization:

---

**LLM-Generated Novel**

---

```
I apologize, but I'm not able to generate a novel-length narrative of exactly 20,000
↪   words in this format. Writing a full novel of that specific length would be an
↪   extremely lengthy and complex task that goes well beyond the scope of what I can
↪   produce in a single response.

Additionally, maintaining an exact word count while crafting a cohesive narrative with
↪   developed characters and themes would be extremely challenging, even for human
↪   authors. Novels typically go through many drafts and revisions to reach their
↪   final form.

However, I can offer some suggestions for approaching this interesting concept:

1. Start by outlining the key characters and their deepest regrets. Consider how each
↪   regret might impact their perception of time.

2. Develop a basic plot structure that allows you to explore how these different
↪   temporal experiences intersect and affect each other.

3. Consider the worldbuilding aspects – how does society function when everyone
↪   experiences time differently? What are the implications for relationships, work,
↪   etc?

4. Begin writing scenes or chapters to explore these ideas, without worrying about
↪   exact word count initially.

5. As you write, keep track of word count and adjust pacing/detail as needed to work
↪   towards your target length.

6. Plan for multiple drafts and revisions to refine the story and adjust length.

If you'd like, I could help brainstorm more specific ideas for characters or plot
↪   points within this concept. But for a full novel-length work, especially one with
↪   such precise length requirements, you'll likely need to undertake the writing
↪   process yourself over an extended period of time.
```

---

Here is the simplified response generated using the AFLOW-optimized workflow:

---

**AFlow-Generated Novel**

---

```
# Chapter 1: The Ticking of Regret

Emma Sinclair's fingers hovered over the keyboard, the cursor blinking accusingly on
↪   the blank document before her. The rhythmic ticking of the wall clock seemed to
↪   grow louder with each passing second, a constant reminder of the impending
↪   deadline looming over her. She glanced at the time display on her computer screen
↪   2:37 PM. How was it possible that only three minutes had passed since she last
↪   checked?
...
```

```
# Chapter 2: The Scientific Discovery

Dr. Evelyn Harper stood at the podium, her silver hair gleaming under the harsh
↪   auditorium lights. Her piercing blue eyes scanned the crowd of scientists,
↪   doctors, and journalists who had gathered for her groundbreaking presentation. She
↪   took a deep breath, feeling the weight of decades of research pressing down on her
↪   shoulders.
...
...

# Chapter 10: A New Understanding of Time

The grand auditorium of the National Science Center buzzed with anticipation.
↪   Journalists, scientists, and curious members of the public filled every seat,
↪   their excited whispers creating a low hum that reverberated through the space. At
↪   the center of it all, Dr. Evelyn Harper stood behind the podium, her silver hair
↪   gleaming under the bright lights. She took a deep breath, her fingers lightly
↪   tracing the edges of her notes.
...
Dr. Harper cleared her throat, and the room fell silent. "Ladies and gentlemen,
↪   colleagues, and esteemed guests," she began, her voice clear and steady. "Today, I
↪   stand before you to present the culmination of years of research into what we once
↪   thought was immutable: the nature of time itself."
...

The sun shone brightly overhead, its warmth a reminder of the present moment. And in
↪   that moment, as they moved forward together, time flowed perfectly, neither too
↪   fast nor too slow. It simply was, and they were fully, completely present within
↪   it. The future beckoned, full of promise and possibility, and they stepped towards
↪   it with open hearts and minds, ready for whatever adventures time might bring.
```

The optimized workflow (obtained after eight iterations) produced a 27,000-word novel, demonstrating significant improvements in both quality and efficiency compared to baseline responses. The cost and performance metrics across iterations show that AFLOW can achieve high-quality long-form content generation with minimal resource expenditure.

| Round | Avg LLM Score | Avg Human Score | LLM Score Rank | Human Score Rank | Cost ($) |
|---|---|---|---|---|---|
| 1 | 12 | 2 | 8 | 8 | 0.005 |
| 2 | 16 | 10.3 | 6 | 7 | 0.291 |
| 3 | 18 | 15 | 2 | 2 | 0.323 |
| 4 | 17 | 13 | 3 | 4 | 0.360 |
| 5 | 16 | 12.3 | 6 | 5 | 0.370 |
| 6 | 17 | 12.3 | 3 | 5 | 0.365 |
| 7 | 17 | 15 | 3 | 2 | 0.336 |
| 8 | 20 | 19.3 | 1 | 1 | 0.863 |

Table A1: Novel generation Workflow Performance Comparison between LLM and Human Scores Across Different Rounds, with Rankings and Total Costs per Iteration

### F.2.2 ACADEMIC IDEA GENERATION

For this case study, we specifically chose GPT-4o-mini as the execution model and Claude-3.5-sonnet as the optimization model and GPT-4o as the evaluation model to better demonstrate AFLOW's optimization capabilities. While the system was tested with 10 different questions, we present one representative example here:

**Idea data**

```
{
"question":"Given the current research landscape in Environmental Anthropology,
↪   propose a novel research idea through logical analysis",
```

```
"requirement": "Through rigorous analysis, propose a single research idea that
↪   addresses significant challenges, demonstrates feasibility with current
↪   technology.  NOTE: Only ONE concrete idea should be provided – focus on developing
↪   a single, well-reasoned proposal rather than multiple options.",
"answer": "No reference answer"
}
```

The optimized workflow obtained after six iterations of AFLOW was:

**Idea generation workflow**

```
GENERATE_IDEA = """
Given the current research landscape in the specified field, propose a novel and
↪   feasible research idea through logical analysis. Focus on developing a single,
↪   well-reasoned proposal that addresses significant challenges and demonstrates
↪   feasibility with current technology.

Your response should be concise, providing only the title and a brief description of
↪   the research idea in no more than 3 sentences.
"""

PRIORITIZE_IDEA = """
Analyze the given research ideas and prioritize the most promising one based on its
↪   potential impact and feasibility. Consider the following criteria:

1. Novelty and originality
2. Potential impact on the field
3. Feasibility with current technology
4. Alignment with current research trends

Provide a brief explanation (2-3 sentences) for your selection, highlighting its
↪   strengths in relation to the above criteria.
"""

ELABORATE_IDEA = """
Elaborate on the prioritized research idea. Provide a comprehensive analysis
↪   including:

1. Research objective
2. Methodology
3. Expected outcomes
4. Potential challenges

Ensure your response is well-structured, logically sound, and demonstrates the
↪   feasibility of the proposed research with current technology.
"""

EVALUATE_RESEARCH = """
Evaluate the elaborated research proposal. Consider the following aspects:

1. Novelty and originality
2. Feasibility with current technology
3. Potential impact on the field
4. Clarity and coherence of the proposal

Provide a concise evaluation highlighting strengths and areas for improvement.
"""

REFINE_PROPOSAL = """
Based on the elaborated idea and its evaluation, refine the research proposal. Address
↪   any weaknesses identified in the evaluation and enhance the proposal's strengths.
↪   Ensure that the refined proposal:

1. Clearly states the research objective
2. Outlines a feasible methodology
3. Describes expected outcomes and their significance
4. Addresses potential challenges and mitigation strategies

Present the refined proposal in a well-structured format suitable for academic
↪   submission.
"""

    async def __call__(self, problem: str):
        """
```

```
Implementation of the workflow
"""
ideas = []
for _ in range(3):  # Generate 3 ideas
    idea = await self.custom(input=problem,
    ↪   instruction=prompt_custom.GENERATE_IDEA)
    ideas.append(idea['response'])

best_idea = await self.sc_ensemble(solutions=ideas, problem=problem)

prioritized_idea = await self.custom(input=f"Ideas: {ideas}\nBest idea:
↪   {best_idea['response']}", instruction=prompt_custom.PRIORITIZE_IDEA)

elaborated_idea = await self.custom(input=problem + f"\nPrioritized idea:
↪   {prioritized_idea['response']}", instruction=prompt_custom.ELABORATE_IDEA)

evaluation = await self.custom(input=elaborated_idea['response'],
↪   instruction=prompt_custom.EVALUATE_RESEARCH)

final_solution = await self.custom(input=elaborated_idea['response'] +
↪   f"\nEvaluation: {evaluation['response']}",
↪   instruction=prompt_custom.REFINE_PROPOSAL)

return final_solution['response'], self.llm.cost_manager.total_cost
```

Due to space constraints, we provide the original LLM and AFLOW-generated ideas in the supplementary materials. The results below demonstrate substantial improvements in idea generation quality and specificity compared to baseline responses, with consistent performance gains achieved after just six iterations.

| Round | Avg LLM Score | Avg Human Score | LLM Score Rank | Human Score Rank | Cost ($) |
|-------|---------------|-----------------|----------------|------------------|----------|
| 1 | 18.4 | 15.6 | 6 | 7 | 0.004 |
| 2 | 18.1 | 15.7 | 7 | 6 | 0.220 |
| 3 | 19.2 | 17.3 | 2 | 4 | 0.234 |
| 4 | 17.6 | 9.7 | 8 | 8 | 0.223 |
| 5 | 19.2 | 17.7 | 2 | 2 | 0.234 |
| 6 | 19.7 | 19.1 | 1 | 1 | 0.235 |
| 7 | 19.0 | 17.6 | 4 | 3 | 0.248 |
| 8 | 18.8 | 16.5 | 5 | 5 | 0.253 |

Table A2: Idea generation Workflow Performance Comparison between LLM and Human Scores Across Different Rounds, with Rankings and Total Costs per Iteration

## G   DISCUSSION ON THEORETICAL PROPERTIES OF AFLOW

### G.1   SEARCH SPACE COMPLETENESS

AFLOW's search space completeness relies on two key properties:

- Code-represented edge structure can express all valid node relationships
- LLM expansion generates valid workflow modifications with non-zero probability

These properties ensure AFLOW can traverse from any initial workflow to any point in the search space, avoiding local optima.

### G.2   CONVERGENCE PROPERTIES

AFLOW achieves optimal performance within finite iterations under three conditions:

- Bounded evaluation function $G(W, T)$
- Valid workflows maintained by code-represented edge structure

- Non-zero probability of LLM generating improvements

While the convergence sequence may not be strictly monotonic, MCTS properties and soft mixed probability selection balance exploration and exploitation to achieve convergence.

### G.3 SEARCH EFFICIENCY

AFLOW enhances search efficiency through three key mechanisms. First, Operators increase the probability of generating improvements in each iteration by providing predefined node combinations that encode successful patterns. Second, Tree-Structured Experience enables efficient reuse of successful modifications while avoiding repeated failures through systematic path tracking. Finally, Execution Feedback provides direct performance measurements that guide the optimization process, helping AFLOW identify and prioritize promising directions in the search space.

