# OpenReview forum: "AFlow: Automating Agentic Workflow Generation"
_ICLR.cc/2025/Conference — ICLR 2025 Oral_

### Official Review · Reviewer_D4oh · 2024-11-03

**Soundness:** 3
**Presentation:** 3
**Contribution:** 3
**Rating:** 8
**Confidence:** 3

**Summary:**

This paper proposes AFlow, a new automated framework that uses Monte Carlo tree search to find the optimal workflow in the exploration space. It also redefines the workflow through code modification, tree-structured experience, and execution feedback. This paper verifies the effectiveness of AFlow on 6 different benchmark datasets, and enables smaller models to outperform GPT-4o on specific tasks. The main contributions of this paper are as follows:

1. Problem Formulation: This paper formalizes the workflow optimization problem and generalizes previous approaches to specific cases. This provides a unified framework for future research at both node and workflow optimization levels.
2. This paper designs AFlow, an MCTS-based method that automatically discovers efficient workflows across multiple domains with minimal human intervention.
3. This article evaluates AFlow on six benchmark datasets: HumanEval, MBPP, MATH, GSM8K, HotPotQA, and DRO, verifying the effectiveness of AFlow. It is worth noting that the workflow generated by AFlow enables smaller LLMs to outperform larger models, providing better cost-effectiveness, which has a significant impact on practical applications.

**Strengths:**

1. Originality

This paper pioneered a new definition of workflow optimization. By redefining the workflow process, it transformed the problem of automatically generating workflows into a Monte Carlo tree search problem in the search space. It also made up for the shortcomings of the previous problem, transformed the previous work into special cases, and provided a unified framework for subsequent researchers.

2. Quality

The AFlow framework proposed in the paper has shown strong performance in experimental evaluation, outperforming existing methods by an average of 5.7% on benchmark datasets. Additionally, AFLOW can achieve better performance than larger models using smaller LLM models, which is of great importance for practical applications.

3. Clarity

The paper provides detailed descriptions and explanations of the key components of the AFLOW framework, such as the MCTS algorithm, node selection strategy, and LLM-driven node expansion, making the entire implementation process clear and understandable. At the same time, the paper also describes the experimental setup and experimental process in great detail, and the explanation of the experimental results is also clear. The entire paper is logically rigorous and well-organized.

4. Significance

The AFlow proposed in this paper not only has a significant improvement in performance, but also proposes a unified framework in the field of automatic generation and optimization of workflows. The new definition better explains this task and provides a new framework and optimization direction for future research.

**Weaknesses:**

This paper has achieved good results in both method and experiment. In terms of method, the paper innovatively formulated the automatic workflow optimization problem, establishing a foundational structure for future research. In terms of experiment, it not only achieved good results, but also conducted a lot of relevant analysis. However, this paper has some weaknesses in the following aspects:

1. The experimental part of this paper lacks the cost analysis of the early AFlow search stage. The cost analysis of different methods later in the paper shows the effectiveness and low consumption of the workflow found by AFlow, but the early MCTS search is a huge process, and the execution of nodes will also consume certain resources. This part does not provide experimental explanation.  If the cost of exploring and finding the optimal workflow is huge, then the discussion on cost should include this resource consumption.
2. This paper argues that different language models require different workflows to achieve their optimal performance. However, there is a lack of sufficient experiments to support this assertion, as the paper only mentions that the workflow identified using DeepSeek-V2.5 performs notably weaker on GPT-4o-mini compared to the workflow found using GPT-4o-mini itself.  At least one more set of comparative experiments should be added, that is, generate a workflow through GPT-4o-mini and then use DeepSeek-V2.5 and GPT-4o-mini respectively to see the experimental results. It would be best if more comparative experiments of other types of models could be added, such as adding another LLama series model, and comparing the three models. This is an interesting assertion, but more sufficient experiments are needed to verify it.

**Questions:**

1.  Can you provide examples of optimized workflows for different tasks, along with explanations of how these workflows can be interpreted.
2. How consistent are the results when running AFlow multiple times with the same model and task? What methods, if any, does AFlow use to introduce variability in the search process? How is the scope of the search space defined and controlled in AFlow?

---

### Official Review · Reviewer_hYzq · 2024-11-03

**Soundness:** 3
**Presentation:** 1
**Contribution:** 3
**Rating:** 6
**Confidence:** 3

**Summary:**

This paper proposes a novel approach to agentic workflow that leverages MCTS to search over sequences of actions that are jointly able to achieve automation of various tasks.

**Strengths:**

The methodology is interesting, and the experimental setup is convincing in showing that AFLow enables smaller models to achieve superior performance to larger models. This lifts the cost/accuracy Pareto front. Given these results I am appreciative of this work.

**Weaknesses:**

While the methodology and results are nice, I do however believe that the presentation of the paper requires improvement. The description of the AFLow methodology lacks precision and can at times even be called handwavy (details below), which makes the paper hard to read. In case authors are able to address those issues I may be willing to increase my score. Some concrete examples (more examples in the questions):
- There is a tree structure involved in the MCTS search process, but there is also a graph and nodes involved in the search space of workflows. This is a potential point of confusion and many parts of the paper do not make explicit which of those two graphs they are talking about.
- The exact definition of the search space is not made clear anywhere in the paper. It is clear that a node is a tuple consisting of a node, prompt, temperature, and output format. What is then less clear is what it means to have an edge between two nodes. Does this mean that we first made the first LLM invocation (of the first node), and then sequentially following that, we make the second LLM invocation (of the second node)? And what does it then mean if our node has two outgoing edges (is this a decision point where we invoke one or the other, or do we execute both next nodes in parallel)? What is also less clear is how the output of the first LLM invocation is used in the LLM calls for later nodes (if at all). Is there a root node? These all point to a broader issue of a lack of precision in the specification of AFlow.

**Questions:**

- Page 3: “We define an agentic workflow W as a sequence of LLM-invoking nodes”. Does this definition of a sequence really make sense? It appears that it can also be some sort of graph structure that allows for loops, decision points (branching), and parallel relations. Terminology-wise, wouldn’t that graph be the workflow, and a sequence of LLM-invocations would then be a particular instantiation or execution of that workflow? This relates also to Page 4: “the goal of workflow optimization is to discover a workflow W …”, given the definition of W as merely a sequence, is this what is intended here?
- Page 4: “The edges E can represent various structures, such as: Graph: A flexible structure representing hierarchical, sequential, or parallel relationships between nodes, allowing for complex branching workflows”. The graph that is shown in Figure 2 is merely a DAG. How can a DAG represent both sequential and parallel relationships in one graph? Typically richer graph languages are required to model rich types of branching workflows, such as workflow nets [1] or BPMN. Is the graph structure that is intended here a DAG and thereby unable to represent parallel relationships, or richer than a DAG and thereby able to represent parallel relationships and other complex workflow patterns [3]? Note that efficient search spaces for have been proposed for some of those workflow representations (e.g., [4]).
- Page 4, about Code: “offering the most precise control over workflow execution”. Why would this be the most precise? Various variants of Petri nets and other graphical representations that are commonly used in the literature to define workflows are turing complete.
- Page 6: Equation 2: “W^* = AFlow(S_{AFlow}, G, T)“. It is unclear what method AFlow here refers to. My thought initially was that it refers to Algorithm 1 “Algorithm of AFlow”. But that algorithm is defined to take 5 arguments (defined in the algorithms require), while equation 2 passes three arguments. Without clarity on this, the procedure is not well specified.
- Page 6: “AFlow can perform searches with an empty Operator Set”. If the set of operators is empty, then what is the set of mutations to the workflow that are considered? I didn’t find a definition of this.
- Page 7: “Our algorithm forms the initial node by evaluating an empty workflow on the validation set, which is distinct from the root node in MCTS”. I did not understand this. Why is the initial node of the workflow not the root node in MCTS?
- Page 7 about expansion: not clear if the LLM generates new workflow only (i.e., new code), or whether it also generates/modifies the contents of the nodes (e.g., the prompt/model/output format).

[1] van der Aalst, W. M.P. (1997). Verification of workflow nets. In: International Conference on Application and Theory of Petri nets (pp. 407-426). Springer.

[2] Dijkman, R. M., Dumas, M., & Ouyang, C. (2008). Semantics and analysis of business process models in BPMN. Information and Software technology, 50(12), 1281-1294.

[3] van der Aalst, W. M.P., Ter Hofstede, A. H., Kiepuszewski, B., & Barros, A. P. (2003). Workflow patterns. Distributed and parallel databases, 14, 5-51.

[4] Esparza, J.: Synthesis rules for Petri nets, and how they lead to new results. In: International Conference on Concurrency Theory. pp. 182–198. Springer (1990)

---

### Official Review · Reviewer_ciCS · 2024-11-03

**Soundness:** 3
**Presentation:** 3
**Contribution:** 4
**Rating:** 8
**Confidence:** 3

**Summary:**

This paper introduces AFLOW, a framework that automates the generation and optimization of agentic workflows for Large Language Models (LLMs) by reformulating workflow optimization as a Monte Carlo Tree Search (MCTS) problem. The workflow structure consists of LLM-invoking nodes connected by edges, represented in code. The system leverages tree-structured experience and execution feedback to refine workflows iteratively. AFLOW is evaluated across six benchmark datasets, demonstrating performance improvements over manual and automated baseline approaches and enabling smaller models to achieve competitive results at lower cost. This novel approach aims to reduce human intervention in workflow design, providing a scalable and efficient framework for workflow optimization in structured tasks.

**Strengths:**

Novel Approach: AFLOW’s integration of MCTS with code-represented workflows introduces a new direction in automating LLM workflows. This reduces the reliance on manual design and allows efficient workflow discovery and optimization.

Comprehensive Problem Formulation: The paper formalizes workflow optimization with a general mathematical framework, effectively unifying prior approaches and broadening the potential for future applications.

Detailed Critique of Prior Work: The authors present an insightful analysis of existing methods, identifying the limitations of prior frameworks like ADAS in handling information accumulation and search efficiency. This sets a strong foundation for AFLOW’s proposed contributions.

Empirical Validation: AFLOW is rigorously evaluated across diverse benchmark tasks, with quantitative comparisons to multiple baselines. Ablation studies further illustrate the impact of different operators, and cost analysis demonstrates AFLOW’s efficiency.

**Weaknesses:**

Limited Scope and Generalizability: The paper primarily demonstrates AFLOW on benchmark tasks with clear success metrics, which raises questions about its applicability to more open-ended tasks, such as document generation or creative exploration. There is limited discussion on how the standardized prompts used in AFLOW would generalize to tasks without clear success criteria. The current prompts seem tailored to test-taking scenarios and may lack the flexibility required for tasks that demand creative or exploratory outputs. A clearer strategy for adapting or evolving these prompts to support open-ended workflows would strengthen the paper’s position on generalizability.

Implementation Details for Reproducibility: While the automated workflow design is a strength, the paper lacks sufficient details on prompt handling, tool calling, and how workflows evolve with execution feedback. Specific examples of workflow changes during optimization and consistency maintenance across components would improve reproducibility.

Limited Comparative Analysis: Although the paper provides a robust critique of ADAS, the treatment of DSPy is brief, and Tree of Thoughts is only briefly mentioned despite its relevance. A more detailed comparison with these approaches would clarify AFLOW’s unique contributions and limitations.

Theoretical Analysis: The paper lacks a theoretical analysis on the convergence of the MCTS optimization process, completeness of the search space, and performance bounds, which are important for understanding the robustness and scalability of the method.

**Questions:**

Scope and Generalizability:
How do you envision AFLOW handling tasks that do not have well-defined success metrics, such as creative writing or exploratory research? Are there specific adaptations you would recommend for such tasks?
The paper primarily discusses AFLOW’s applicability to benchmark tasks with structured goals. Could you provide examples or suggestions for how AFLOW might be applied to open-ended, real-world applications beyond these benchmarks?
Could you clarify how AFLOW maintains context and coherence in longer workflows or workflows requiring memory across stages? Is there a mechanism in place to support long-term contextual awareness?

Prompt Adaptation and Handling for Broader Tasks:
Given that AFLOW’s prompt templates seem tailored to structured problem-solving tasks, what modifications would be necessary to adapt these prompts to open-ended or creative tasks?
Are there mechanisms for adapting or evolving prompts dynamically based on task progress? How does AFLOW approach prompt optimization in situations where task objectives may evolve or remain undefined?
Could you provide examples of prompt templates used in AFLOW, and explain how execution feedback specifically informs prompt revisions, if at all?

Implementation Details for Reproducibility:
Could you provide a clearer example of how workflows evolve iteratively during AFLOW’s optimization process? Specifically, how does feedback inform changes in node structure, prompt design, or operator selection?
How does AFLOW ensure consistency and reliability across different task types, especially when tasks involve diverse workflows or require different prompts and operators?

Comparison with Other Approaches:
How does AFLOW’s MCTS-based approach compare to DSPy’s instruction optimization or the Tree of Thoughts approach in handling complex, multi-stage tasks? Are there specific advantages or limitations relative to these frameworks?
Tree of Thoughts is briefly mentioned as related work but not fully explored. Could you elaborate on how AFLOW’s workflow optimization extends or diverges from the principles underlying Tree of Thoughts?

---

### Official Review · Reviewer_ZL3t · 2024-11-05

**Soundness:** 3
**Presentation:** 3
**Contribution:** 4
**Rating:** 8
**Confidence:** 3

**Summary:**

The paper introduces Aflow, a new automatic workflow optimization framework based on MCTS and code-represented workflows. It models the workflow as a sequence of LLM-invoking nodes, where nodes represent LLM action and edges represent logic, dependencies and flows between the actions. The experiment results on 6 benchmarks show preliminary effectiveness over SOTA, and that AFlow can enable smaller LLMs to outperform larger models, offering better cost-performance efficiency, with significant implications for real-world applications.

**Strengths:**

Overall, this paper is clear, well-motivated and provides a new framework for automatic workflow optimization, which has significant potential impact on agent design and workflow optimization for the broader machine learning community. It proposes a novel, original approach to model the workflow as a sequence of LLM-invoking nodes in a graph structure, with prompts, operators, and code-represented edges in the search space. By leveraging MCTS, the paper reaches SOTA performance on major workflow benchmarks and shows the potential of enabling smaller, cheaper models reaching similar performances as large models. The documentations of the experiment setup, code representation, case studies and results are clear and technically sound, and this paper can provide great inspiration for other researchers in the domain of agent workflow optmization.

**Weaknesses:**

The paper could benefit from discussions with regards to the following points:
1. To reduce the search space, the paper focuses on custom prompts, operators and code-represented edges by fixing parameters such as model choice, temperature and output format - which is a sound choice. Could there be more discussion on the potential effect of these parameters on model performance?
2. The authors mention some of the parameters used in MCTS in the appendix (e.g. $\lambda = 0.4$ used to balance exploration vs. exploitation), but not in the main paper. It would be helpful to include key parameter values and brief discussions about the choices.
3. Similarly, a quick discussion around why models are chosen for specific parts (executor vs. optimizer) would be helpful for context as well.

**Questions:**

The authors chose 6 agentic workflow benchmarks for the experiments. Are there more rationale and explanation behind how those benchmarks are chosen to best represent the agent workflow optimization capability?

---

### Meta-Review · Area_Chair_UYJ9 · 2024-12-20

**Metareview:**

This paper proposes a framework for automatically discovering and optimizing agentic workflows mainly for coding and math problems. The authors have also shown the potential for adopting workflow for open-ended tasks. The framework not only improves on standard benchmarks (code generation, math reasoning, question answering) but also that the discovered workflows enable smaller, less expensive models to achieve performance levels comparable or superior to much larger models. The potential for future expansion further enhances the paper’s value.

All reviewers rate positively on this paper.  During the rebuttal, reviewers acknowledge that major concerns have been addressed. Therefore, the AC recommends accepting this paper.

For the camera-ready version, the authors should carefully revise the paper according to the reviewers' advice, including clarification of workflow representation and execution, generalizability to open-ended tasks and limitations, and model-specific optimal workflows.

**Additional Comments On Reviewer Discussion:**

During the rebuttal and discussion period, the reviewers raised several major concerns about the paper’s clarity, generalizability, and cost analysis. For example:
- *Reviewer ZL3t*, *Reviewer hYzq* and *Reviewer ciCS* requested clearer explanations of the difference between the MCTS search tree and the code-based workflows, how edges and nodes are defined, and how prompts and operators evolve during the search.
- *Reviewers ciCS*, *Reviewer hYzq* and *Reviewer D4oh* inquired about extending the approach beyond tasks with numerical feedback to more creative, open-ended problems.
-  *Reviewer D4oh* and *Reviewer ciCS* requested more details on the cost of the search stage, particularly the expense of the MCTS-driven workflow discovery process.
- *Reviewer D4oh* and *Reviewer ZL3t* wanted stronger evidence that different models require distinct optimized workflows.

To address these concerns, the authors have provided detailed explanations and additional experiments.  Reviewers acknowledged that the major concerns had been solved.

---

### Decision · Program_Chairs · 2025-01-22

Accept (Oral)